**MABEL photon-counting laser altimetry data in Alaska for ICESat-2 simulations**
**and development**
**K. M. Brunt[1,2], T. A. Neumann[2], J. M. Amundson[3], J. L. Kavanaugh[4],**
**M. S. Moussavi[5,6], K. M. Walsh[2,7], W. B. Cook[2], and T. Markus[2]**
[1] {Earth System Science Interdisciplinary Center, University of Maryland, College
Park, Maryland}
[2] {NASA Goddard Space Flight Center, Greenbelt, Maryland}
[3] {University of Alaska Southeast, Juneau, Alaska}
[4] {University of Alberta, Edmonton, Alberta, Canada}
[5] {Cooperative Institute for Research in Environmental Sciences (CIRES), University
of Colorado, Boulder, Colorado}
[6] {National Snow and Ice Data Center (NSIDC), CIRES, University of Colorado,
Boulder, Colorado}
[7] {Stinger Ghaffarian Technologies, Inc., Greenbelt, Maryland}
*Correspondence to:* K.M. Brunt (kelly.m.brunt@nasa.gov)
**Abstract**
Ice, Cloud, and land Elevation Satellite-2 (ICESat-2) is scheduled to launch in late 2017
and will carry the Advanced Topographic Laser Altimeter System (ATLAS), which is a
photon-counting laser altimeter and represents a new approach to satellite determination
of surface elevation. Given the new technology of ATLAS, an airborne instrument, the
Multiple Altimeter Beam Experimental Lidar (MABEL), was developed to provide data
needed for satellite-algorithm development and ICESat-2 error analysis. MABEL was
deployed out of Fairbanks, Alaska in July 2014 to provide a test data set for algorithm
development in summer conditions with water saturated snow and ice surfaces. Here we
compare MABEL lidar data to *in situ* observations in Southeast Alaska to assess
instrument performance in summer conditions and in the presence of glacier surface melt
ponds and a wet snowpack. Results indicate that: 1) based on MABEL and *in situ* data
comparisons, the ATLAS 90 m beam-spacing strategy will provide a valid assessment of
across-track slope that is consistent with shallow slopes (<1°) of an ice-sheet interior over
50 to 150 m length scales; 2) the dense along-track sampling strategy of photon counting
systems can provide crevasse detail; and 3) MABEL 532 nm wavelength light may
sample both the surface and subsurface of shallow (approximately 2 m deep) supraglacial
melt ponds. The data associated with crevasses and melt ponds indicate the potential
ICESat-2 will have for the study of mountain and other small glaciers.
**1      Introduction**
Ice, Cloud, and land Elevation Satellite-2 (ICESat-2) is a NASA mission scheduled to
launch in 2017. ICESat-2 is a follow-on mission to ICESat (2003-2009) and will extend
the time series of elevation-change measurements aimed at estimating the contribution of
polar ice sheets to eustatic sea level rise. ICESat-2 will carry the Advanced Topographic
Laser Altimeter System (ATLAS), which uses a different surface detection strategy than
the instrument onboard ICESat. Abdalati et al. (2010) provides an early overview of the
ATLAS concept and overall design. While the measurement goals of ATLAS remain as
described in Abdalati et al. (2010), some of the details have evolved (Markus et al.,
submitted). ATLAS is a 6-beam, photon-counting laser altimeter (Fig. 1). In a photon-
counting system, single-photon sensitive detectors are used to record arrival time of any
detected photon. ATLAS will use short (< 2 ns) 532 nm (green) wavelength laser pulses,
with a 10 kHz repetition rate, which yields a ~0.70 m along-track sampling interval, and a
~17 m diameter footprint. An accurate assessment of ice-sheet surface-elevation change
based on altimetry is dependent upon knowledge of local slope (Zwally et al., 2011).
Therefore, the six ATLAS beams are arranged into three sets of pairs. Spacing between
the three pair sets is ~3 km to increase sampling density, while spacing between each
beam within a given pair will be ~90 m to make the critical determination of local slope
on each pass. Therefore, elevation change can be determined from only two passes of a
given area (Brunt et al., 2014).
Given this new approach to satellite surface elevation measurement, an airborne
instrument, the Multiple Altimeter Beam Experimental Lidar (MABEL), was developed
to: 1) enable the development of ICESat-2 geophysical algorithms prior to launch; and 2)
enable ICESat-2 error analysis. MABEL (discussed in detail in McGill et al., 2013) is a
multibeam, photon-counting lidar, sampling at both 532 (green) and 1064 (near infrared)
nm wavelengths using short (~1.5 ns) laser pulses. The dual wavelength instrument
design was intended to assess green-wavelength light penetration in water or snow
(McGill et al., 2013). Deems et al. (2013) provides a review of lidar use for snow studies
and describes how light at 532 and 1064 nm wavelengths interacts with snow surfaces.
Light penetration into a snow surface is a function of both grain size (with larger snow-
grain size resulting in increased volumetric scattering, and therefore increased light
penetration) and wavelength (with 532 nm light having lower absorption than 1064 nm
light, which ultimately produces increased light penetration at the shorter wavelength).
Deems et al. (2013) also note that light penetration into snow surfaces is extremely
difficult to accurately measure.
Following engineering test flights in 2010 and 2011, MABEL was deployed to Greenland
(April 2012) and Alaska (July 2014) to collect data, including from glacier targets, and to
assess elements of the resulting data that may vary seasonally. The Greenland 2012
campaign sampled winter-like conditions, while the Alaska 2014 campaign was timed to
collect data during the summer melt season, which is characterized by open crevasses and
surface melt ponds. In winter, increased albedo, reduced ice-sheet surface roughness, and
reduced solar background and backscatter in the atmosphere all lead to an increased
signal-to-noise ratio and an increase in photon-retrieval density (*i.e.*, the number of, and
temporal distribution of photons transmitted and recorded by the lidar). In general, with
increased photon-retrieval density, we expect better surface measurement precision. In
the extreme case, the photon-retrieval density may be sufficiently high that the instrument
receiver does not have the time required to process the incoming photon information
before receiving more. This effect is referred to as 'instrument dead time' and can
produce a positive surface elevation bias. In summer, reduced albedo, increased ice-sheet
surface roughness, and increased solar background leads to a decrease in photon-retrieval
density and signal-to-noise ratios, compromising measurement precision. The Alaska
2014 campaign also aimed to investigate how light at 532 and 1064 nm wavelengths
interacts with the surface in melting conditions, and how this may affect the statistics of
the 532 nm signal photons and overall elevation accuracy.
Here, we compare *in situ* measurements with MABEL airborne lidar data on the Bagley
(16 July 2014; 60.5° N, 141.7° W) and Juneau (31 July 2014; 58.6° N, 134.2° W)
icefields in Southeast Alaska (Fig. 2). These comparisons are made with consideration for
the planned ATLAS beam geometry in order to investigate instrument performance in
summer conditions and in the presence of surface crevasses and melt ponds.
**2      Data and methods**
**2.1    MABEL data**
MABEL data (Level 2A, release 9) for the Alaska 2014 campaign (Fig. 2) are available
from              the              NASA              ICESat-2             website
(http://icesat.gsfc.nasa.gov/icesat2/data/mabel/mabel_docs). Each data file contains 1
minute of data for every available beam (approximately two beams per deployment were
compromised due to instrumentation issues). The data files contain photon arrival times
resulting from reflected laser light (*i.e.*, signal photons), solar background and backscatter
in the atmosphere (*i.e.*, background photons) and, to a lesser degree, detector noise (*i.e.*,
noise photons). A histogram-based surface-finding algorithm developed at NASA
Goddard Space Flight Center (GSFC) was used to discriminate signal photons from
background and noise photons. The algorithm is based on histograms of photon arrival
times in 25 m along-track segments and 10 m vertical bins and assumes a random
distribution of background photons and a symmetric return pulse. Further details of this
surface-finding algorithm are described in Brunt et al. (2014). The GSFC algorithm is
applicable to a wide range of surface types, while most ICESat-2 standard data product
algorithms are surface-type specific (*e.g.*, glacier, sea ice, ocean, vegetation, etc.) and
more rigorous with respect to returns identified as surface signal. The derived surface
elevations are reported relative to the WGS84 ellipsoid.
The MABEL laser pulse repetition rate is variable (5 to 25 kHz) and was 5 kHz for the
data presented here. At this nominal altitude and repetition rate, and at an aircraft speed
of ~200 m s$^{-1}$, MABEL samples a ~2 m footprint every ~0.04 m along-track.
MABEL beams are arranged approximately linearly, perpendicular to the direction of
flight, with the 1064 nm beams leading the 523 nm beams by ~60 m. The system allows
for beam-geometry changes between flights with a maximum beam spread of ~2 km
given the 20 km nominal altitude of the NASA ER-2 aircraft. The beam configuration for
the Alaska 2014 campaign had total swath width of 200 m (Fig. 1). The spacing between
the individual beams was configured to allow simulation of the planned beam geometry
of ATLAS. Previous results from the MABEL 2012 Greenland campaign suggest that the
ATLAS beam geometry is appropriate for the determination of slope on ~90 m across-
track length scales, a measurement that will be fundamental to accounting for the effects
of local surface slope from the ice-sheet surface-elevation change derived from ATLAS
(Brunt et al., 2014).
Relative to one another, the MABEL beams have non-uniform average transmit energy.
While all beams originate from a single 1064 nm laser source, each beam follows a
unique optical path through the instrument once split from the source beam. Several
individual beams maintain the fundamental 1064 nm wavelength of the source, while
others are split off of a beam that is frequency-doubled to 532 nm (McGill et al., 2013).
Owing to the frequency-doubling process and the non-uniform optical paths (fiber
lengths) through the instrument, the 1064 nm and 523 nm transmit-pulse energies are
generally not equal. During the 2014 Alaska campaign, there were fifteen 532 nm beams
and six 1064 nm beams.
Our analysis used relatively high-energy beams. For analyses intended to mimic the 90 m
spacing of the ATLAS beam geometry, two 1064 nm beams were chosen based on their
across-track ground separation and along-track signal-photon density: beams 43 (center
of the array) and 48 (~90 m to the left of the array center across-track). For analyses
intended to assess issues that might be wavelength-dependent, beams 5 (532 nm) and 50
(1064 nm) were chosen because they were in line with one another in the along-track
direction and approximately 35 m across-track to the left of the array center.

Because of the different optical paths each beam takes through the instrument, each MABEL beam has a unique range bias (McGill et al., 2013). Prior to Level 2A data processing, MABEL ranges are corrected for these channel-specific optical path lengths using a calibration derived from data recorded during aircraft pitch and roll maneuvers performed over stretches of open ocean. We assume that this calibration mitigates the larger channel biases, including those associated with errors in pointing. However, other smaller-scale channel biases may still exist; these smaller-scale channel bias corrections were on the order of decimeters. Much of the analysis performed here, such as evaluation of local surface slope, did not require absolute range accuracy. Therefore, the individual beams were generally only calibrated with respect to one another based on data collected over the nearest flat surface (*e.g.*, open water). These calibrations were made relative to the beam closest to the center of the array.

## 2.2 MABEL camera imagery

For the 2014 Alaska campaign, a camera was integrated with MABEL and was successful for over 40% of the campaign's duration. The images were typically used to visually confirm the type of surface being overflown by MABEL (*e.g.*, ice, open water, sea ice, or melt ponds) or to confirm the presence or absence of clouds. These images are also available on the ICESat-2 website. The MABEL camera (Sony Nex7, with a 55 to 220 mm, f/4.5-6.6 telephoto lens) was mounted on the same optical bench as the MABEL telescopes and shared the same portal in the aircraft. For the 2014 Alaska campaign, a focal length of 210 mm was used for the duration of the campaign. The camera produced 6000 by 4000 pixel color images. At a nominal aircraft altitude of 20 km, each image covers an approximately 2.25 by 1.5 km area, or approximately 3 m per pixel at sea level. Images were taken every 3 seconds, which provided approximately 30% overlap between images. The images collected were not systematically georeferenced; however, they were time-stamped based on MABEL instrument timing to provide a first-order assessment of the surface that the lidar had surveyed.

## 2.3 Landsat 8 and WorldView-2 imagery

Data from the Landsat 8 Operational Land Imager (OLI) of the Bagley Icefield (Fig. 2b) were used as an independent assessment of the depths of melt ponds surveyed by MABEL. We applied spectrally based depth retrieval models to Landsat 8 imagery (Moussavi et al., 2016; Moussavi, 2015; Pope et al., 2015), which were calibrated based on data from supraglacial lakes in Greenland. We assessed the performance of OLI's coastal blue, blue, green, red, and panchromatic channels in retrieving supraglacial lake depths. Ultimately, the models establish a relationship between Landsat 8 top-of-atmosphere (TOA) comparing pre-drainage spectral reflectance values over the lakes with a post-drainage digital elevation model (DEM), derived from WorldView-2 imagery acquired from the Polar Geospatial Center at the University of Minnesota, using image-processing software (ERDAS). Our analysis indicated that for shallow lakes (depth < 5 m), red and panchromatic band data are most suitable for supraglacial bathymetry. Because of the relatively small size of the lakes in our study area, we chose the panchromatic channel for the better spatial resolution.

A second WorldView-2-derived DEM was used near the terminus of the Lower Taku Glacier (Fig. 2c) to assess surface elevations derived from MABEL signal photons in steep and crevassed terrain. The DEM, created by the Polar Geospatial Center at the University of Minnesota, was extracted from high-resolution along-track stereo WorldView-2 imagery processed with NASA's open source Ames Stereo Pipeline software (Moratto et al., 2010). The WorldView-2 images were collected on 6 June 2014, while the MABEL data were collected on 16 July 2014, and thus separated by 40 days. As part of an unrelated project, GPS data were continuously collected at six sites on the Lower Taku Glacier throughout the summer, using a Trimble NetR9 receiver; these data were used to tie the MABEL survey data to the WorldView-2 DEM. The data were processed kinematically using the Plate Boundary Observatory station AB50, located at the Mendenhall Glacier Visitor Center, approximately 20 km west of the survey area.

## 2.4    Juneau Icefield GPS data

Previous studies (Brunt et al., 2013; Brunt et al., 2014) have demonstrated that MABEL precisely characterizes the ice-sheet surface when comparing MABEL-derived slope on

90 m across-track length scales with those based on both Airborne Topographic Mapper
(ATM; Krabill et al., 2002) and Laser Vegetation Imaging Sensor (LVIS, more recently
referred to as Land Vegetation Ice Sensor; Blair et al., 1999).
We conducted a GPS survey on the Juneau Icefield (Fig. 2c) to determine the length-
scale at which a ground-based local slope assessment on a flat surface (<1° slope) begins
to differ significantly from that of a 90 m across-track slope assessment. On 19 July
2014, we conducted differential GPS surveys of the nodes of a series of concentric
equilateral triangles. WGS84 ellipsoidal heights, in a Universal Transverse Mercator map
projection (UTM zone 8N), were determined for each node using Trimble 5700 base and
rover receivers, operating in real-time differential mode. The base-station receiver was
located at the Juneau Icefield Research Program (JIRP) Camp 10, approximately 1 km
from where the rover receivers were operated. Eight triangles were surveyed with side
lengths of 5, 10, 25, 50, 75, 90, 125, and 150 m (Fig. 3, black points). We fit a surface to
each of the eight triangles and then calculated the surface slope in both the UTM easting
and northing directions (surface gradients $\delta z/\delta x$ and $\delta z/\delta y$).
MABEL-based surface gradients $\delta z/\delta x$ and $\delta z/\delta y$ were generated from data from the 31
July 2014 flight and compared with the GPS-based surface gradients. We used beams 43
and 48 (1064 nm), which had relatively high along-track signal-photon density and
approximately 90 m ground spacing, and intersected the GPS survey array (Fig. 3, red
lines). The MABEL beams were cross-calibrated to remove the relative elevation bias
resulting from their different optical paths through the instrument. To accomplish this
calibration, we chose beam 43 as a reference beam, calculated the mean difference
between the elevation of the signal photons of the reference beam and beam 48 over the
nearest open ocean, and removed that offset (0.2 m) from beam 48. We assumed that the
calibration remained valid for the 75 km between the open ocean and the GPS survey
area. We projected the geodetic MABEL data to the gridded map projection of the GPS
data (UTM zone 8N) to facilitate direct comparisons and so that changes in elevation in
both the easting and northing directions (surface gradients $\delta z/\delta x$ and $\delta z/\delta y$) could be
treated uniformly. We generated a MABEL triangle, with nodes based on the
intersections of the GPS survey and the ground tracks of the MABEL beams (Fig. 3, blue
solid points). Elevations at those nodes were determined by taking an average of the
elevations of the signal photons within a 5 m radius of those points, to take into account
MABEL horizontal geolocation uncertainty. We then fit a 1 m by 1 m gridded surface to
those points and calculated the associated MABEL surface gradient in both the easting
and northing directions ($\delta z/\delta x$ and $\delta z/\delta y$). Based on this surface, the local slope for the
survey area was determined to be 0.5°, or comparable to what we expect for an ice-sheet
interior. Finally, we generated a surface based on the three GPS survey sites that were
closest to the nodes that defined the MABEL surface (Fig. 3, blue open circles).
We compared the MABEL-derived slopes to the slopes from each of the concentric GPS
triangles and the slope based on the GPS survey sites that were closest to the nodes that
defined the MABEL surface. Specifically, we created a surface gradient comparison
(SGC) parameter for each of the GPS-derived triangles (*i*) by calculating the square root
of the sum of the squares (RSS) of the differences between the MABEL-derived and
GPS-derived slopes in both the easting and northing (x and y) directions:
$$SGC_{(i)} = \sqrt{\left[(\delta z/\delta x)_{MABEL} - (\delta z/\delta x)_{GPS_{(i)}}\right]^2 + \left[(\delta z/\delta y)_{MABEL} - (\delta z/\delta y)_{GPS_{(i)}}\right]^2}, \qquad (1)$$
where $\delta z/\delta x$ and $\delta z/\delta y$ are the surface gradients associated with both MABEL and each of
the GPS triangles (*i*), in the easting and northing directions.
**3      Results**
**3.1     MABEL signal-photon density**
For illustrative purposes, we produced histograms of the MABEL surface-return for the
beams used in our analyses (Fig. 4; beams 5, 43, 48, and 50) from 3000 m of along-track
data over a stretch of open ocean. We calibrated the beam elevations to one another to
remove the unique beam elevation biases; relative bias corrections ranged from 0.03 to
0.73 m. We then detrended the surface elevations based on a linear fit to the signal
photons to remove any elevation differences associated with wind stress or the relatively
small effects of ocean dynamic topography and geoid undulation. The detrending of each
beam takes into account all of these effects; this correction ranged from 0.11 to 0.29 m
over the 3000 m of along-track data used for this analysis. We then produced histograms
using a 0.01 m vertical bin size. We determined the full width at half maximum (FWHM)
for each of the beams, which ranged from 0.19 m in beam 5 (532 nm) to 0.31 m in beam
43 (1064 nm). From Fig. 4, the relative differences in the signal strengths of the
individual beams are evident from the non-uniform amplitudes of the photon-count
distribution.
The MABEL signal often has a primary surface return and a second, weaker return
approximately 0.5 to 1.5 m below the surface. This is due to unintended secondary pulses
from the MABEL laser that occur under some operational conditions. The exact
conditions for after-pulsing are not completely understood, but are most likely the result
of temperature drifts in the fundamental laser system. These occur due to changing
environmental conditions within the instrument pod in the aircraft, and/or changes in
efficiency of the coolant system. The cooling system relies upon passive external fins
exposed to ambient cold conditions at altitude and these conditions (temperature, airflow)
change during flight. The secondary laser pulses are primarily seen in the 1064 nm
returns, and are minimized when the 1064 nm source is frequency-doubled to generate
532 nm beams. This second pulse can affect statistics associated with MABEL results
and was therefore manually removed. This secondary pulse is evident in the open-ocean
data example at approximately 0.75 m below the main surface return (Fig. 4).
Given nearly uniform surface conditions, along-track signal-photon density for each
beam varied within and between flights based on parameters such as weather conditions,
time of day, and sun-incidence angle. The signal-photon densities on the Juneau and
Bagley icefields, for each beam considered here, are given in Table 1. These densities are
reported based on 0.70 m along-track length scales for direct comparison with previous
results (Brunt et al., 2014), to mimic the ATLAS sampling interval (one laser shot every
0.70 m). MABEL along-track signal-photon densities for the July 2014 Alaska campaign
were lower than those reported during the April 2012 Greenland campaign by Brunt et al.
(2014). They reported 3.4 and 3.9 signal photons per 0.70 m for beams 5 and 6 (532 nm),
respectively, over the Greenland Ice Sheet; the highest counts of signal photons per 0.70
m were 1.8 and 3.7 for 532 and 1064 nm channels, respectively (Table 1). Some of this
variation may have been related to seasonal differences in surface reflectivity between the
two campaigns, which include parameters such as the freshness of the most recent
snowfall, the dust content of the surface, the presence (or absence) of surface melt and
ponds, and the presence (or absence) of snow bridges that cover crevasses. Some
variation may also have been related to instrumentation issues, such as cleanliness of the
elements in the optics.
The MABEL signal-photon densities (Table 1) are generally lower than those expected
for ATLAS. Under similar conditions as the 2014 MABEL summer campaign and based
on performance models, we expect the strong beams of ATLAS to record 7.6 signal
photons every shot (or 0.70 m along track) over ice sheets and 0.5 to 1.8 signal photons
every shot over the open ocean, dependent upon the state of the wind (A. Martino, NASA
GSFC, personal communication 2016). We note that for the Alaskan icefields, the
expected number of signal photons based on the performance model is probably too high,
as the model uses an albedo of 0.9, which is more appropriate for ice with fresh snow or
the interior of Antarctica than for ice fields in Alaska in summer. Relative to the
performance model, at best (*i.e.*, using data from beam 50) the MABEL data used in this
analysis suggest that the signal-photon densities were ~72% of the expected ATLAS
signal-photon densities over open ocean (with calm winds) and ~49% of the expected
ATLAS signal-photon densities over summer ice sheets.
**3.2    Elevation bias and uncertainty**
We compared MABEL elevations to those based on the Juneau Icefield GPS array,
interpolated to the MABEL/GPS points of intersection (Fig. 3, blue solid points). The
mean offset, or bias, for the three points of intersection was 3.2 ±0.1 m. While this ~3 m
instrument bias is larger than that of other airborne lidars, it is within the MABEL design
goals (algorithm development and error analysis), where instrument precision is more
critical to satellite algorithm development than absolute accuracy. Thus, while other
photon-counting systems are being used for change detection (*e.g.*, Young et al., 2015),
in its current configuration, MABEL is not suitable for time-series analysis of elevation
change, either independently or when integrated with other datasets.
We assessed the surface precision of MABEL data (*i.e.*, the spread of the MABEL data
point cloud about a known surface, or the standard deviation of the mean difference
between MABEL and a known surface elevation, Hodgson and Bresnahan, 2004) over
the flat stretch of open ocean used in the analysis of Fig. 4. For approximately 3000 m of
along-track open water, the surface-precision estimates for the strong 532 and 1064 nm
beams, based on a standard deviations of the mean differences from the detrended
surface, were ±0.11 and ±0.12 m, respectively. Brunt et al. (2014) reported similar
surface-precision values (±0.14 m) based on direct comparison of MABEL elevation data
with high-resolution ground-based GPS data (differentially post-processed with an RMS
< 0.05 m) over an airport departure apron. Further, Brunt et al. (2013) reported that for all
MABEL campaigns between 2010 and 2014 for which similar ground-based GPS data
were available, MABEL surface precision ranged between ±0.11 and ±0.24 m. During
that time period, MABEL had been deployed on two different types of aircraft and in a
number of different optical configurations (McGill et al., 2013). These return pulse
widths are dominated by the width of the MABEL transmit pulse (~1.5 ns) and show
relatively little pulse broadening due to surface slope or roughness.
## 3.3 Surface characterization
We examined MABEL data from the Bagley and Juneau icefields and from the Lower
Taku Glacier to determine how well photon-counting laser altimeters would capture
surface detail on relatively short length scales (less than 1 km), such as crevasses and
melt ponds.
Analysis of data from individual beams over the Bagley Icefield indicates that MABEL
can capture surface detail of crevasse fields. Fig. 5a shows stitched MABEL images of
one set of crevasses on the Bagley Icefield; Fig. 5b shows MABEL signal and
background photons for a 1200 m range that includes the glacier surface; and Fig. 5c
shows MABEL signal photons, indicating returns from both the glacier surface and the
bottoms of a series of crevasses. The along-track slope of this crevasse field, between
140.60° and 140.58° W longitude in Fig. 5c, is 1°.
Similarly, analysis of the individual beams in a different area of the Bagley Icefield
indicated that MABEL can determine the location of melt ponds. Fig. 6a shows stitched
MABEL images from crevasse and melt-pond fields on the Bagley Icefield; Fig. 6b
shows MABEL signal and background photons for a 1200 m range window that includes
the glacier surface; Fig. 6c shows both signal and background photon-count densities (per
125 shots, or ~2.5 m of along-track distance); and Fig. 6d shows MABEL signal photons,
indicating the location of two melt ponds, which are approximately 50 and 70 m in along-
track length. The along-track slope of this crevasse and melt pond field, between 141.91°
and 141.93° W longitude in Fig. 6d, is 0.5°. A histogram of the signal photons associated
with the larger melt pond in Fig. 6d is provided in Fig. 7. This figure depicts how light at
532 and 1064 nm wavelengths interacts with the surface of the melt pond, and how the
melt pond affects the statistics of the 532 nm return signal. The FWHM for the 532 and
1064 nm return signal were 0.26 and 0.34 m, respectively. From Figs. 6 and 7 we observe
that while no distinct features corresponding to the bottoms of the melt ponds are visible,
an increased spread is apparent in the 532-nm histogram, likely associated with
volumetric scattering throughout the ponds. We applied spectrally based depth-retrieval
models to Landsat 8 imagery (Moussavi et al., 2016; Moussavi, 2015; Pope et al., 2015)
for an independent assessment of the depth of the melt-pond on the Bagley Icefield in
Fig. 6d. This analysis indicated that melt ponds in this region were approximately 2 m
deep.
Analysis of data from individual beams near the terminus of the Lower Taku Glacier
(Fig. 8) demonstrates MABEL performance in regions with steeper slopes. The slope in
this region is 4°, and is similar to slopes near ice-sheet margins; this slope also
corresponds to the maximum slope angle used for ATLAS performance modeling over
ice-sheet margins (A. Martino, NASA GSFC, personal communication 2014). Fig. 8a
shows stitched MABEL camera images, which suggest a much rougher surface than that
of the low slope areas of interest on the Bagley Icefield examined in Fig. 6. Additionally,
the MABEL ice-surface signal near the terminus was slightly compromised due to
intermittent cloud cover, which attenuated the MABEL transmitted and/or received laser
pulses. Further, when cloud cover allows for only intermittent surface determination, the

surface-finding algorithm used to discriminate signal photons from background and noise photons is compromised.

MABEL-derived surface elevations over the Lower Taku Glacier were compared to elevations from the WorldView-2-derived DEM (Fig. 8b), which had 2 m horizontal-resolution. Fig. 8c is one of the images used to create the DEM shown in Fig. 8d. The MABEL data were collected 40 days after the WorldView-2 images were acquired. GPS data from the Lower Taku Glacier were used to determine mean ice-flow velocities to tie the two datasets together. Specifically, the MABEL ground tracks were migrated up ice flow, using the northing and easting components of the mean velocities derived from the GPS data, to more accurately compare MABEL surface elevations to those derived from the earlier WorldView-2 imagery. An elevation was then extracted from the WorldView-2 DEM for each migrated MABEL data point.

Mean ice-flow velocities varied substantially for the sites on the Lower Taku Glacier (Fig. 8c). A mean ice-flow velocity of 0.2 m day$^{-1}$ was recorded at the southern GPS site (SDWN, 800 m from the center of the MABEL ground track), while mean velocities for the two central GPS sites (C1 and SLFT, 1500 m from the center of the MABEL data ground track) were 0.7 m day$^{-1}$ and mean velocities for the three northern GPS sites (C2, SRIT, and SUP, 3000 m from the center of the MABEL data line) were 1.0 m day$^{-1}$. While the flow velocity at SDWN does not necessarily represent flow along the entire MABEL data line, we chose this GPS site for data migration purposes based on proximity to the center of the data line and because the direction of flow in the northing and easting directions matched the southern end of the MABEL data line. MABEL elevations were 8 m ±2.5 m lower than the values extracted from the WorldView-2 DEM. This bias is higher than other biases assessed during this campaign, which we attribute to: 1) the difference between the WorldView-2 DEM elevation and true elevation, which can be on the order of meters when uncorrected (Shean et al., 2016); 2) the 3 m MABEL range bias, determined over the open ocean (Fig. 4); and 3) the amount of surface melting that occurred between June and July, which was assessed at the GPS sites to be 2.3 m using ablation wires. Further, we note that elevation uncertainty is a function of MABEL horizontal uncertainty (2 m) and surface slope; therefore, steeper terrain leads to greater overall elevation uncertainty (Brunt et al., 2014).

## 3.4    Slope assessments

Using Eq. (1), we compared the MABEL-derived surface-gradient comparison (SGC) parameters to those based on the Juneau Icefield GPS array (Fig. 9). The MABEL-derived SGC parameters were consistent with GPS-derived SGC parameters over length scales ranging from 50 m (just over half of the ATLAS beam spacing) to 150 m (just under twice the ATLAS beam spacing). The SGCs for 50 to 150 m spatial scales were less than 0.5°.

The high-resolution WorldView-2 DEM also provided a means of assessing MABEL-derived across-track slopes in steeper glacial settings. Using a method similar to that of Brunt et al. (2014), we calculated a ~40 m across-track MABEL-derived slope and compared this with a ~40 m across-track slope based on WorldView-2 DEM elevations. The MABEL-derived across-track slope was calculated using beams 43 and 50, migrated to match the timing of the WorldView-2 image acquisition and limited to continuous stretches of the southern part of the data line (Fig. 8b). Along-track signal-photon density for beam 48 was insufficient to allow for a 90 m across-track assessment. The MABEL data from each beam were aligned to determine along-track elevations and across track slopes between beams. A DEM-derived across-track slope was calculated based on elevations that were extracted from the DEM at each interpolated MABEL data point for beams 43 and 50. Fig. 10a shows a comparison between the MABEL and DEM elevations associated with beam 43, while Fig. 10b shows a comparison between MABEL-derived and DEM-derived across-track slopes. The total along-track distance used in this analysis was ~300 m (see box in Fig. 8b). The mean residual between the MABEL-derived slope and the DEM-derived slope was 0.25°.

## 4    Discussion

As noted above, there are some significant differences between MABEL and ATLAS depicted in Fig. 1 (*e.g.*, number of beams, beam pattern, and altitude) and described elsewhere in this paper (*e.g.*, footprint size, along track footprint spacing, and

wavelengths). In order to relate the predicted performance of ATLAS with the measured performance of MABEL, some common metric is necessary that accounts for as many of the differences as is practicable. The signal-photon density is a metric to relate the radiometry of the two instruments. Given that the signal-photon density is generally less than that predicted for ATLAS, for a given background rate, the surface should be more easily distinguished in ATLAS data. While in theory one could use the framework developed for predicting ATLAS radiometric characteristics to make similar predictions for MABEL and therefore use MABEL data to evaluate that framework, the efficiency or radiometric throughput of MABEL has not been characterized well enough to do so. Flight data (Brunt et al., 2014) show that for a given campaign, the measured signal-photon density of MABEL changes by tens of percent over relatively uniform ice sheet interior. Similar changes are measured for the background rate, after consideration for sun angle is taken into account. As such, the analysis presented here cannot be used to quantitatively assess the likelihood that ATLAS will meet its measurement requirements (or the mission science objectives). What we can say is that if the ATLAS signal-photon density and signal-to-noise ratios are within 30% of its measurement requirements (and thus mimics the MABEL performance documented in this study), ATLAS can be used to measure surface slopes over both relatively flat ice-sheet interior conditions and steeper glaciers such as the Lower Taku Glacier, and identify melt ponds. If ATLAS fully meets its measurement requirements, we expect that the definition of small-scale surface features such as crevasses and melt ponds will be correspondingly improved.

The result of this analysis indicates that the MABEL-derived local slope assessment, on a relatively flat glacial surface and on a 90 m across-track length scale, is consistent with *in situ* slope assessments made at spatial scales ranging from 50 to 150 m. For a planar surface where slope is less than 1°, such as the interior of an ice sheet, we expect the local slope measured by a GPS survey and MABEL to be similar over a wide range of spatial scales. Any small differences observed between the two survey techniques would likely reflect 1) the non-planarity of the surface and/or 2) the sensitivity of the results to small-scale slopes or roughness captured by one measurement technique and not the other. With the good observed agreement between MABEL-derived and GPS-derived slope assessments over 50-150 m length scales (Fig. 9), we are confident that the ATLAS 90 m

beam-spacing strategy will provide a non-aliased estimate of local slope for ice-sheet
interiors (<1°) over these spatial scales. This knowledge is necessary for accurate
assessments of ice-sheet surface-elevation change.
Based on our comparison with a WorldView-2-derived DEM of the Lower Taku Glacier,
MABEL can also provide valid estimates of across-track slope, even in steeper terrain.
Once migrated for GPS-derived ice-flow displacements, the southern part of the
MABEL-derived surface elevations are in good agreement with the DEM data, and the
slope comparison between MABEL-derived and DEM-derived across-track slopes had a
mean residual of 0.25°. This residual is larger than that reported over the Greenland Ice
Sheet (<0.05°) by Brunt et al. (2014), a difference that we attribute to errors associated
with the migration of the MABEL data (and the result of a flight line that was oblique to
the local direction of ice flow). Since the GPS array on the Lower Taku Glacier was not
optimized to facilitate an across-track slope comparison similar to the comparison made
higher up on the Juneau Icefield (Figs. 3 and 9), we do not expect as close an agreement
between the two methods of estimating across-track slope.
Figs. 5c and 6d suggest that the dense along-track sampling of MABEL is sufficient to
capture surface detail, including melt-pond information, from a single, static beam in
regions of low slope, consistent with that of an ice-sheet interior. Based on the continuous
nature of the surface return through the crevasse field, especially in the 1064 nm beam
(50) in Fig. 5c, we conclude that MABEL frequently retrieves a signal from the bottom of
crevasses. Further, Fig. 8b indicates that MABEL continues to provide surface detail in
regions of steeper slope, including the retrieval of the steep slopes of the crevasse walls
(*e.g.*, Figs 5c and 6d).
As previously noted, MABEL data used in this analysis had signal-photon densities that
are ~44% of the expected ATLAS signal-photon densities over summer ice sheets (A.
Martino, NASA GSFC, personal communication 2014). Therefore, we believe that the
level of detail that will be provided by ATLAS will be sufficient to determine local
surface characteristics, similar to those observed on the Lower Taku Glacier. Such
knowledge is critical to determining ice-sheet surface-elevation change, as features that
could compromise these calculations (such as deep crevasses) can move or advect with
ice-sheet flow or be bridged seasonally and must therefore be identifiable in the ATLAS
data.
The crevasse characterization we performed on the Bagley Icefield is qualitatively
confirmed using the camera imagery (Fig. 5a). However, it should be noted that we have
no means of quantitatively assessing the accuracy of MABEL-derived crevasse depths.
Crevasses on an ice-sheet surface have an influence on albedo (Pfeffer and Bretherton,
1987). This variation in reflectance is evident in Figs. 5b, 6b, and 6c, where MABEL
background photon counts, and the signal-to-noise ratios, change significantly. Changes
in MABEL background photon densities have also been used to detect leads in sea ice
(Kwok et al., 2014; Farrell et al., 2015). From Fig. 6c we note that the overall background
photon counts decrease significantly over the eastern region of this plot, which is
characterized by crevasses; however, this change is non-uniform. Background photon
counts drop steadily to nearly zero over the two melt ponds surveyed along this transect.
Penetration of 532 nm wavelength light into the surface, be it a melt pond or snow, is an
ongoing area of research for ICESat-2 algorithm development. MABEL geolocation
uncertainty, and the fact that the 1064 and 532 nm beams do not have coincident
footprints for more direct comparison (as the 1064 nm beams lead the 532 nm beams by
~60 m), compromised our ability to further interrogate this topic with this dataset, as the
data could not be precisely co-registered spatially. Due to these limitations, a separate
campaign with a different photon-counting laser altimeter (with both a more accurate
geopositioning system and coincident 1064 and 532 nm footprints) was deployed to
Thule, Greenland, in July and August 2015 (Brunt et al., 2015). Processing and analysis
of that dataset are still ongoing.
Analysis of MABEL data over small melt ponds on the Bagley Icefield in Alaska
provided a preliminary assessment of how green-wavelength photon-counting systems
will interact with water on an ice surface. Based on the signal-photon elevations in Fig.
6d, and the histogram of the signal photons in Fig. 7, the total spread of the signal
photons, at a wavelength of 532 nm, is approximately 1.5 to 2 m. Further, analysis of
Landsat 8 and WorldView-2 imagery confirm that the melt ponds in this region are
approximately 2 m deep. These results suggest that, while there isn't a distinct signal

return from a melt-pond bottom, the 532 nm MABEL beam may be sampling the entire melt-pond water column. The 1064 nm MABEL beam shows evidence of a secondary return 1.5 m below the main signal return, due to unintended secondary pulses from the MABEL laser that occur under some operational conditions, and is likely not due to melt-pond bottom returns.

Based on the surface characterization results of MABEL data from the Juneau and Bagley icefields, and the dense, six-beam sampling strategy of ATLAS, we are confident that ICESat-2 will contribute significantly to glacier studies at local and regional scales and in polar and mid-latitudes. While previous studies using satellite laser altimetry have investigated the vertical dimension of rifts in the ice sheet (*e.g.*, Fricker et al., 2005), those studies have been limited to major ice-shelf rift systems, as opposed to smaller-scale crevasses. The 0.70 m along-track sampling density of each individual ATLAS beam is well suited for similar vertical dimension studies, but at finer length-scales, such as those associated with alpine glacier crevasse fields.

## 5    Conclusions

Knowledge of local slope and local surface character are required to accurately determine ice-sheet surface-elevation change. The ATLAS beam geometry includes pairs of beams separated at 90 m across track to enable the determination of local slope in one pass, and therefore to enable the determination of ice-sheet surface-elevation change in just two passes. Based on the analysis of MABEL, ground-based GPS data, and the resultant surface gradient comparison (SGC), we conclude that the ATLAS 90 m beam-spacing strategy will provide a valid assessment of local slope that is consistent with the slope of an ice-sheet interior (<1°) on 50 to 150 m length scales. The density of along-track photon-counting lidar data is sufficient to characterize the ice-sheet surface in detail, including small-scale features such as crevasses and melt ponds. This information is required for accurate determination of ice-sheet surface-elevation change. The dense along-track sampling interval and narrow across-track beam spacing of ATLAS will provide a level of detail of mountain glaciers that has previously not been achieved from satellite laser altimetry. While studies of mountain glaciers stand to benefit greatly from

ICESat-2 data, great care will need to be taken in the interpretation of elevation change of
a heterogeneous surface, such as that associated with crevasses or melt ponds.
The MABEL 2014 Alaska campaign was timed to collect data during the summer melt
season to specifically investigate how 532 nm wavelength laser light interacts with a
melting snow surface. Results from MABEL, and confirmed through analysis of Landsat
8 imagery, suggest that 532 nm wavelength light is likely reflecting from the surface and
subsurface of the 2 m deep supraglacial melt ponds on the Bagley Icefield. This is an
ongoing area of research for ATLAS and ICESat-2 algorithm development.
**Acknowledgements**
Funding for this project was through the NASA ICESat-2 Project Science Office.
Funding for J.M. Amundson was provided by NSF-PLR 1303895. We acknowledge the
considerable efforts of the Project, Science, and Instrument teams of NASA's ICESat-2
and MABEL missions. We thank: Eugenia De Marco (ASRC Aerospace Corp.,
NASA/GSFC) and Dan Reed (Sigma Space Corp., NASA/GSFC) for MABEL
instrument support; Scott Luthcke (NASA/GSFC), David Hancock (NASA/WFF), and
Jeff Lee (NASA/WFF) for MABEL data calibration; Scott McGee and Ya' Shonti
Bridgers (JIRP) for GPS field data collection and data processing support; and
NASA/AFRC (specifically ER-2 pilots Tim Williams and Denis Steele) for Alaska
airborne support. WorldView imagery was provided by the Polar Geospatial Center at the
University of Minnesota, which is supported by NSF-PLR 1043681. GPS receivers for
the survey of the terminus of the Lower Taku Glacier were provided by UNAVCO. GPS
receivers for the JIRP survey were provided by Werner Stempfhuber of the Beuth
Hochschule for Technik University of Applied Sciences. And finally, we thank two
anonymous reviewers for their highly constructive suggestions.

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

**Table 1:** MABEL along-track signal photon densities over the open ocean and the Juneau
and Bagley icefields.

| Beam | MABEL surface-signal photons per 0.70 m | | |
|------|---------------------------------------|----------------|-----------------|
| | open ocean | Juneau Icefield | Bagley Icefield |
| 5 (532 nm) | 0.3 | 1.8 | 1.7 |
| 43 (1064 nm) | 1.2 | 3.5 | 2.8 |
| 48 (1064 nm) | 0.5 | 1.5 | 1.0 |
| 50 (1064 nm) | 1.3 | 3.7 | 3.0 |
| ATLAS [1] | 0.5 – 1.8 [2] | 7.6 [3] | 7.6 [3] |

[1] ATLAS instrument allocated performance.
[2] Dependent upon the wind state: 0.5 for high winds and 1.8 for low winds.
[3] This value is for summer conditions on an ice sheet; we note that for summer ice sheets,
the ATLAS performance model uses an albedo of 0.9, which is more appropriate for ice
with fresh snow or the interior of Antarctica.

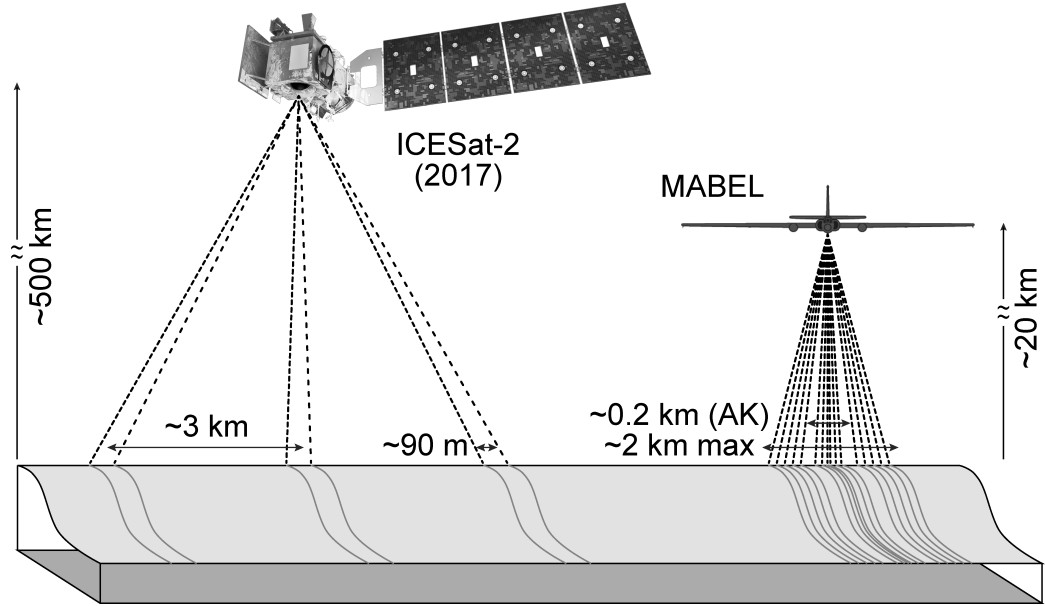

**Figure 1:** Schematic ICESat-2 and MABEL beam geometry (dashed lines) and reference
ground tracks (grey lines along ice-sheet surface). MABEL allows for beam-geometry
changes with a maximum ground spacing of ~2 km at 20 km, however for the 2014 AK
deployment, the maximum ground spacing was 0.2 km (after Brunt et al., 2014).

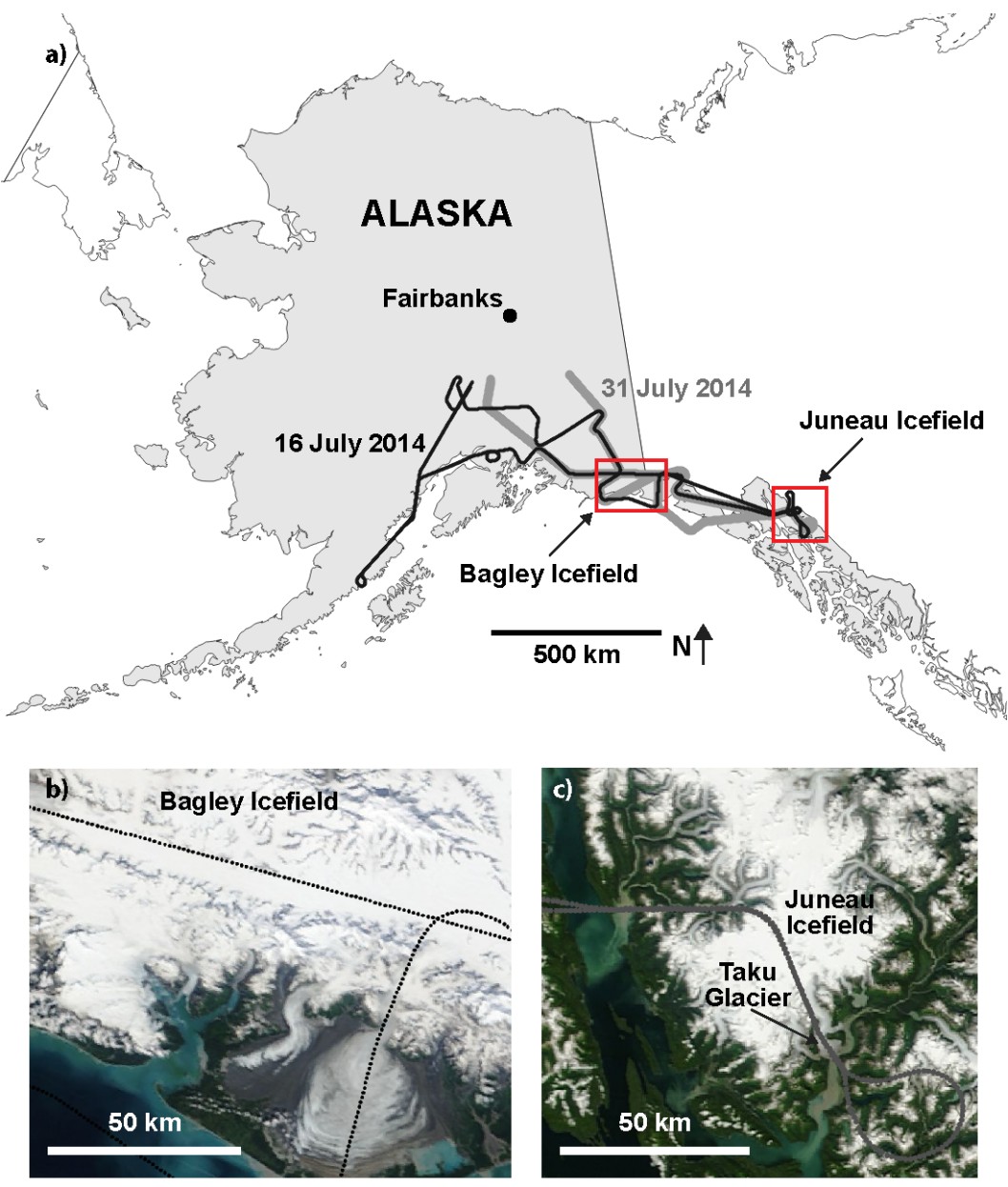

**Figure 2:** Map of the Multiple Altimeter Beam Experimental Lidar (MABEL) flights
used in this analysis from the July 2014 field campaign, which was based out of Fort
Wainwright, Fairbanks, Alaska. **(a)** Overview map, indicating the 16 and 31 July 2014
flight paths. **(b)** Inset of the Bagley Icefield, showing the 16 July 2014 flight path. **(c)**
Inset of the Juneau Icefield, showing the 31 July 2014 flight path and the Taku Glacier.
Both insets are shown with 31 July 2104 MODIS imagery.

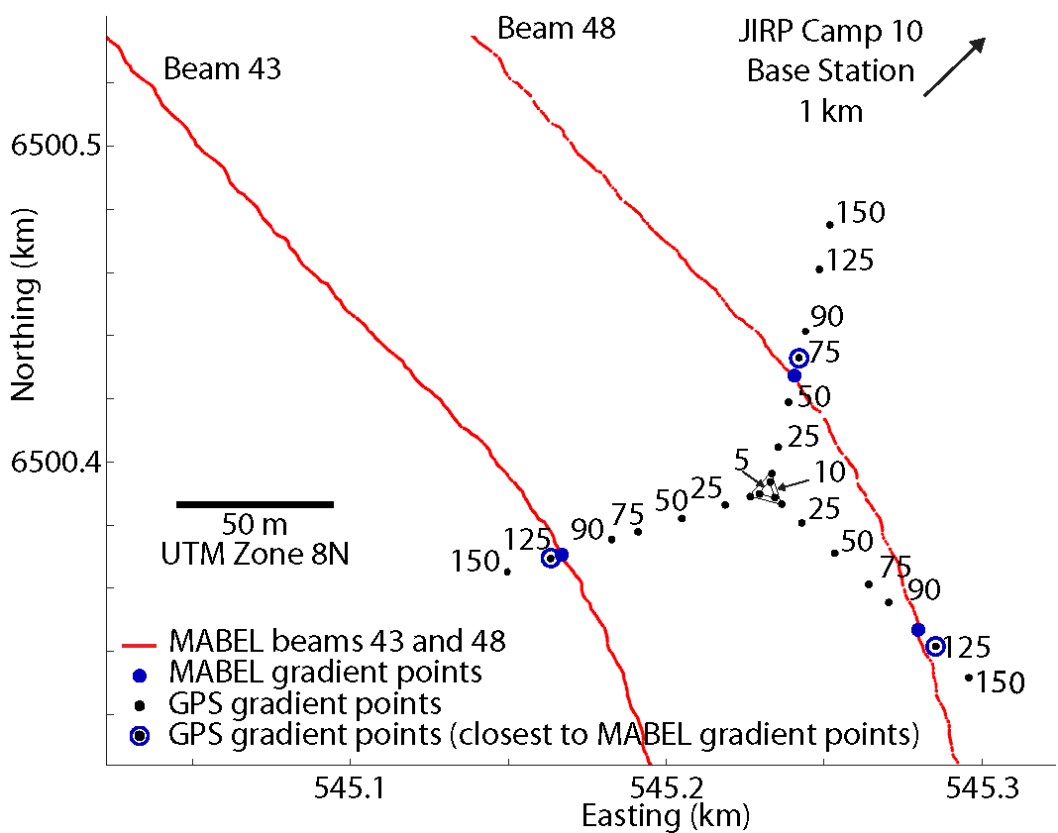

**Figure 3:** GPS survey on the Juneau Icefield. Ground tracks for MABEL beams 43 and 48, from the 31 July 2014 flight, are indicated (red lines). GPS survey points of the nodes of concentric, equilateral triangles, with side lengths of 5, 10, 25, 50, 75, 90, 125, and 150 m, are indicated (black points). Also indicated are the intersections of the MABEL flight lines with the GPS survey grid (blue solid points), which were used to calculate MABEL surface gradients ($\delta z/\delta x$ and $\delta z/\delta y$). The GPS sites that are the closest to the MABEL gradient points are also indicated (blue open circles). The overall slope, based on the MABEL elevations at the points of intersections with the GPS survey grid (blue solid points), is approximately 0.5°.

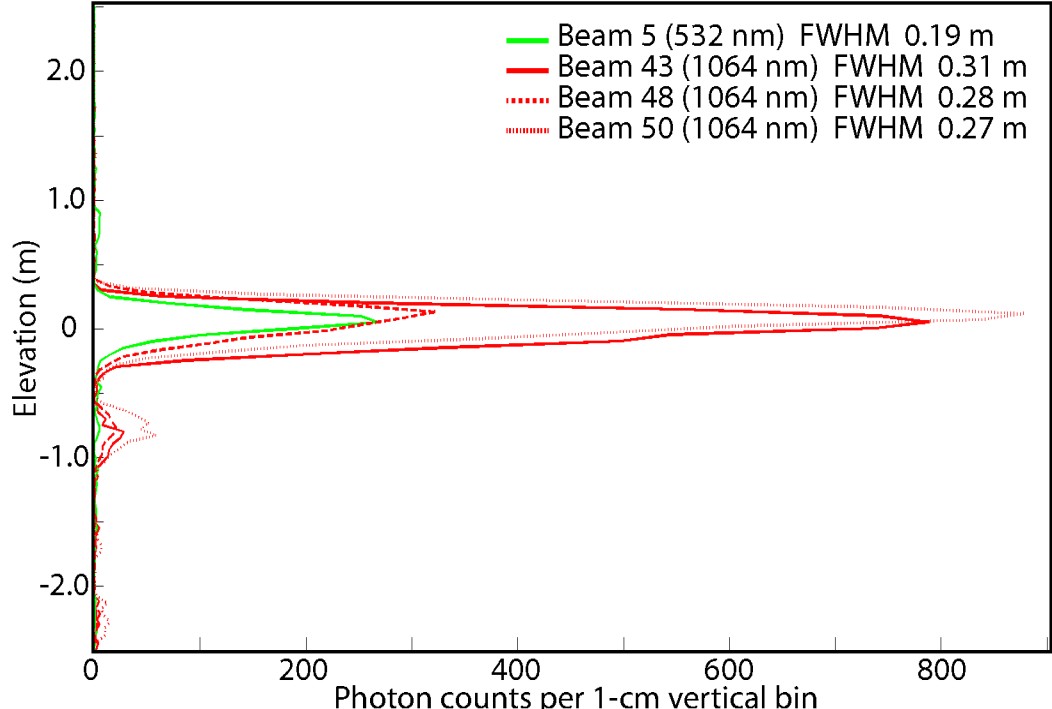

**Figure 4:** Histograms of the signal return for the MABEL beams used in this analysis (5, 43, 48, and 50). Plotted are ocean surface-return photon counts (per 0.01 m vertical bins) over a 3 km along-track distance against elevation (m). The elevations are calibrated to one another and detrended. The full width at half maximum (FWHM) for each histogram are indicated in the legend. The secondary return 0.75 m below the main signal return, which is more evident in the 1064 nm beams, is due to unintended secondary pulses from the MABEL laser that occur under some operational conditions; this was removed for FWHM analysis.

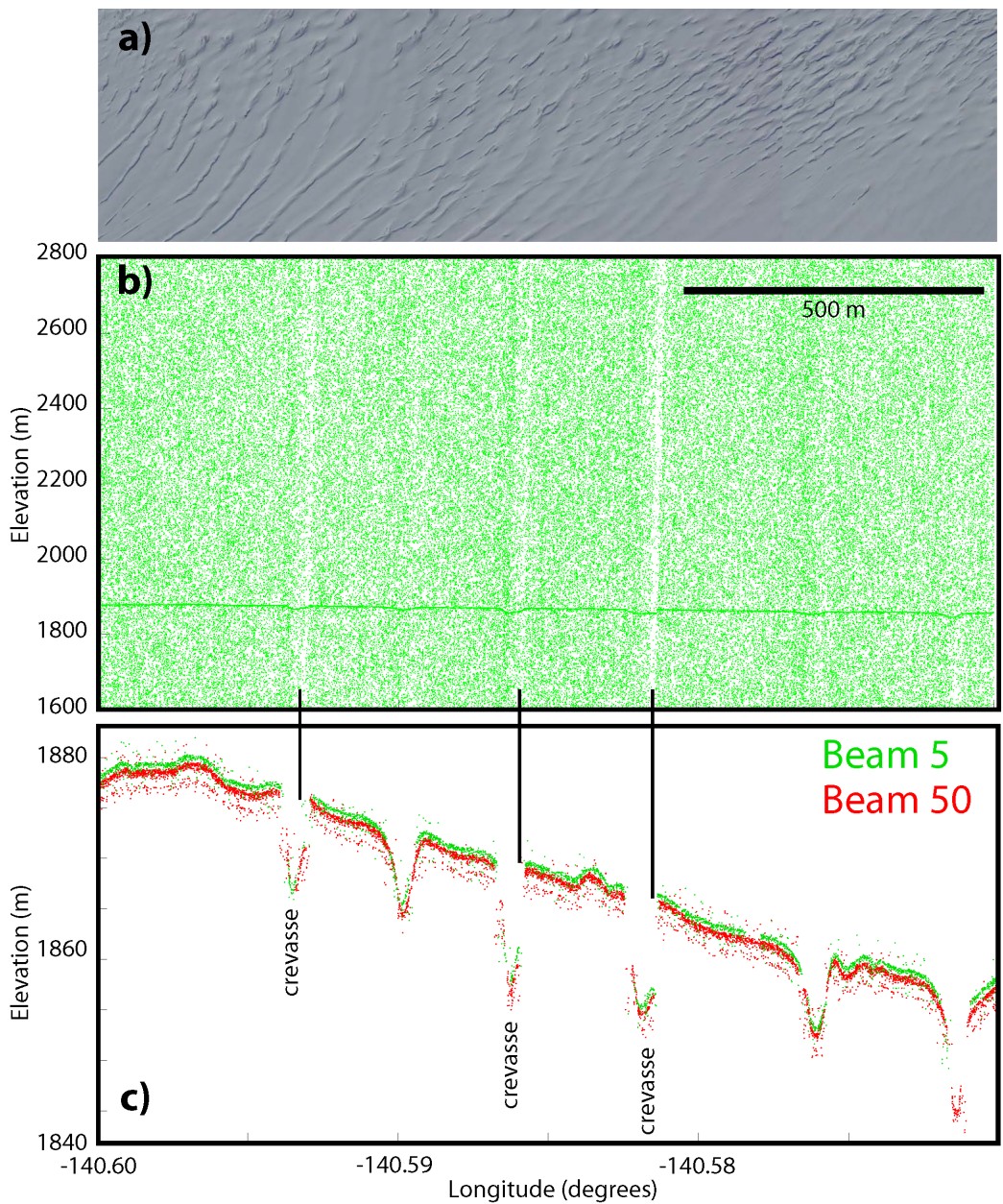

**Figure 5:** MABEL camera and photon data over a heavily crevassed section of the
Bagley Icefield, from the 16 July 2014 flight. **(a)** Stitched MABEL camera images. **(b)**
MABEL signal and background photons for a 1200 m range that includes the glacier
surface. **(c)** MABEL signal photons, indicating both the surface and the bottoms of
crevasses. The along-track slope of this field, between 140.60° and 140.58° W longitude
is 1°.

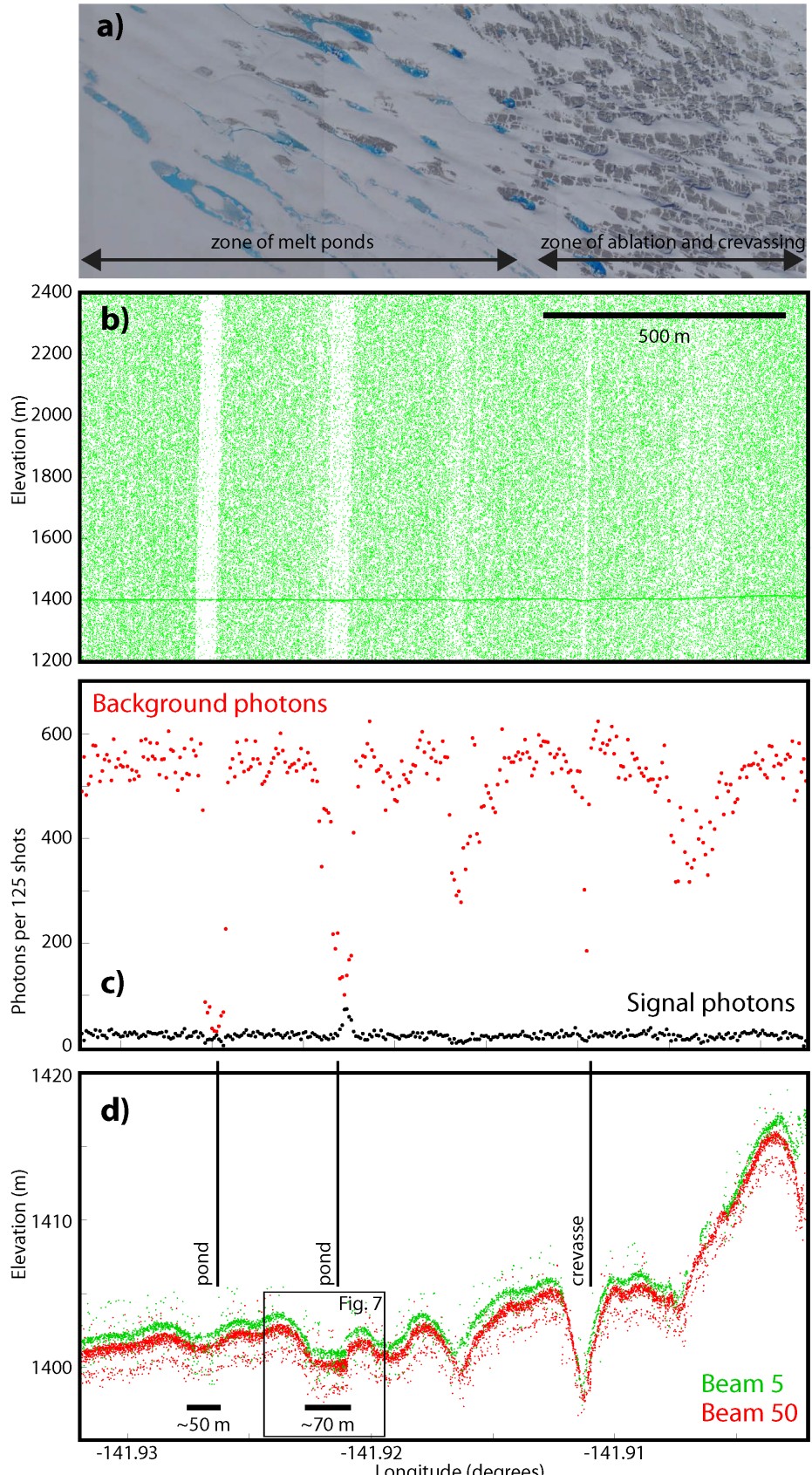

**Figure 6:** MABEL camera and photon data over crevasse and melt-pond fields on the
Bagley Icefield, from the 16 July 2014 flight. **(a)** Stitched MABEL camera images. **(b)**
MABEL signal and background photons for a 1200 m range that includes the glacier
surface. **(c)** Signal (black) and background (red) photon counts per 125 shots
(approximately 2.5 m of along-track distance). **(d)** MABEL signal photons, indicating the
location of melt ponds; the ponds indicated are approximately 50 and 70 m in along-track
length. Fig. 7 is a histogram of the ~70 m pond. The 1064 nm beam shows evidence of a
secondary return 1.5 m below the main signal return, due to unintended secondary pulses
from the MABEL laser that occur under some operational conditions. The along-track
slope of the crevasse field, between 141.93° and 141.91° W longitude is approximately
0.5°.

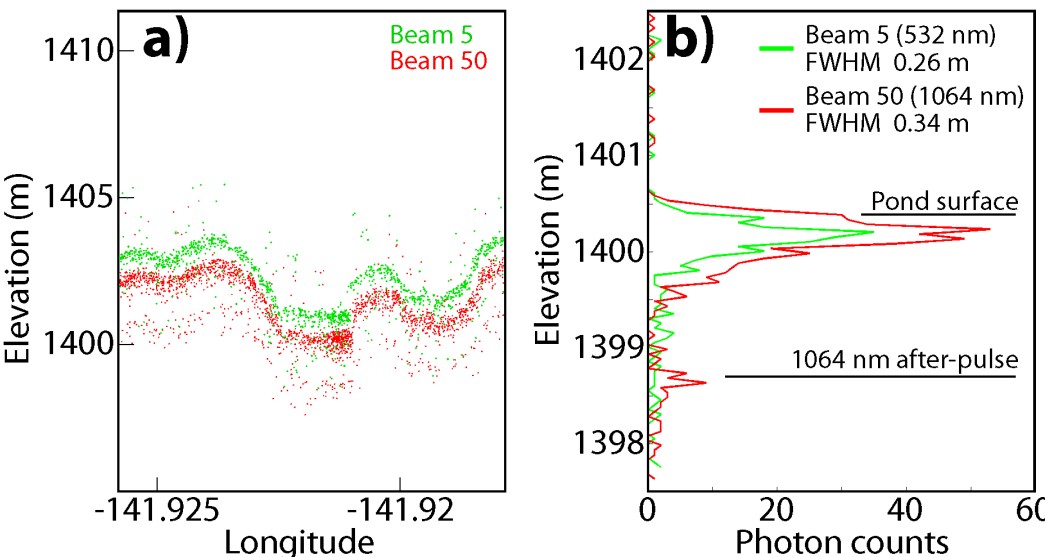

**Figure 7:** Surface return and histogram of the signal return for MABEL beams 5 (532 nm) and 50 (1064 nm) over the ~70 m melt pond in Fig. 6. **(a)** MABEL signal photons over for beams 5 and 50 for the 70 m melt pond in Fig. 6. The 1064 nm beam shows evidence of a secondary return 1.5 m below the main signal return, due to unintended secondary pulses from the MABEL laser that occur under some operational conditions. **(b)** Plotted for each beam are surface-return photon counts per 0.01 m vertical bins against elevation (m). The elevations of beams 5 and 50 are calibrated to one another. The full width at half maximum (FWHM) for each histogram are indicated in the legend. The secondary return <1 m below the main signal return, which is more evident in the 1064 nm beam, is due to unintended secondary pulses from the MABEL laser that occur under some operational conditions; this was removed for FWHM analysis.

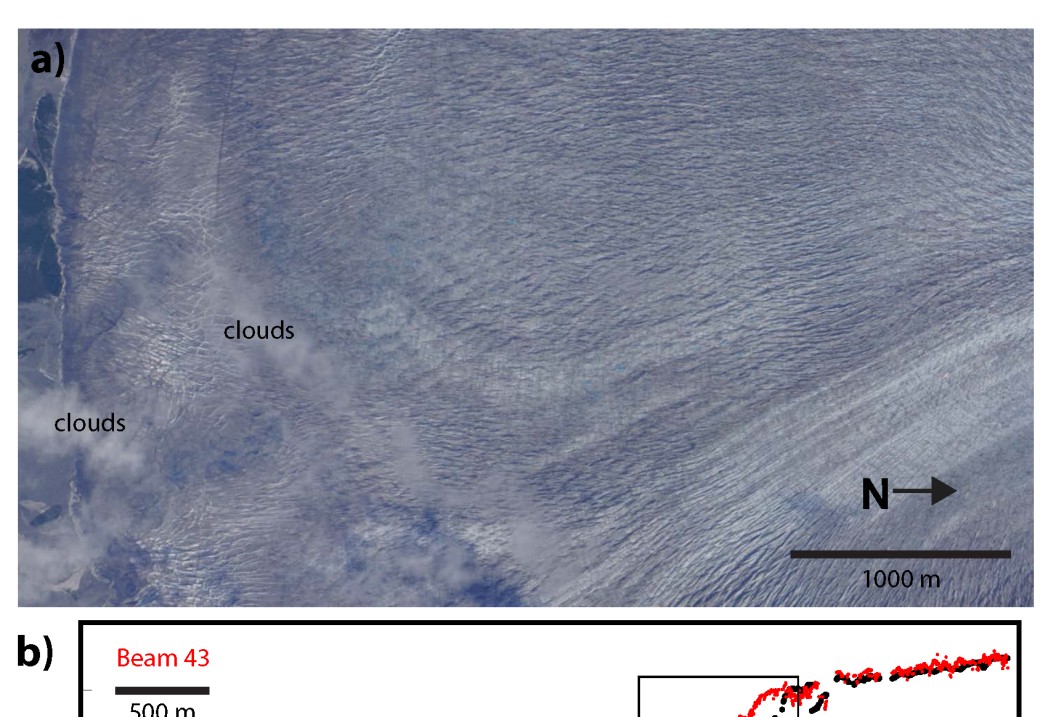

a)

clouds

clouds

N→

1000 m

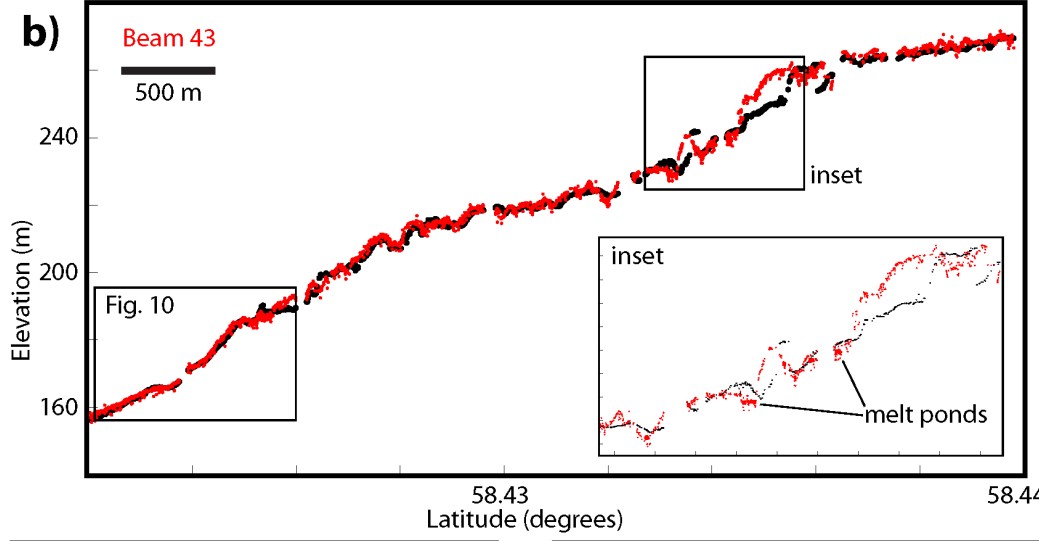

b)

Beam 43

500 m

Elevation (m)

240

200

160

Fig. 10

inset

inset

melt ponds

58.43

58.44

Latitude (degrees)

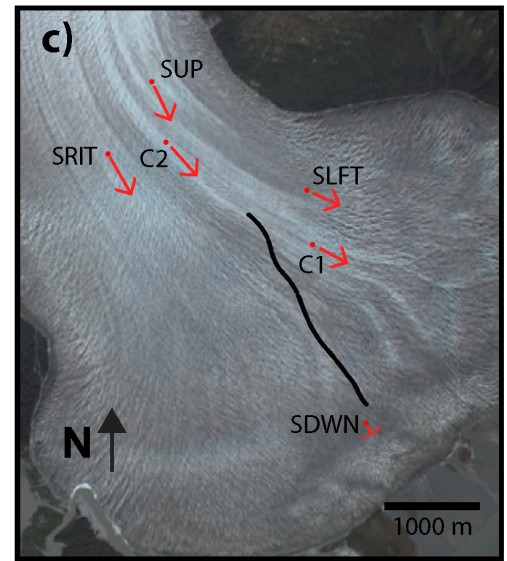

c)

SUP

SRIT

C2

SLFT

C1

SDWN

N

1000 m

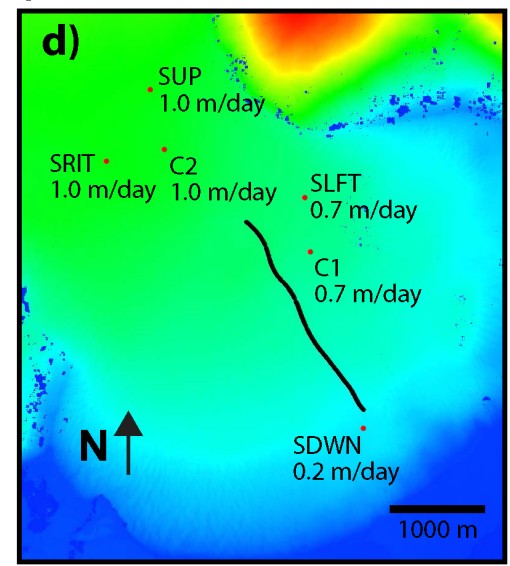

d)

SUP
1.0 m/day

SRIT
1.0 m/day

C2
1.0 m/day

SLFT
0.7 m/day

C1
0.7 m/day

SDWN
0.2 m/day

N

1000 m

**Figure 8:** MABEL data over crevasse fields on the Lower Taku Glacier, from the 16 July
2014 flight. **(a)** Stitched MABEL camera images. **(b)** MABEL signal photons (red),
migrated based on GPS data and corrected for an 8 m range bias, and elevations extracted
from the WorldView-2 DEM (black). **(c)** WorldView-2 image (Copyright DigitalGlobe,
Inc.) with MABEL flight line and GPS sites (red). **(d)** WorldView-2 DEM (Moratto et
al., 2010) with MABEL flight line and GPS sites (red).

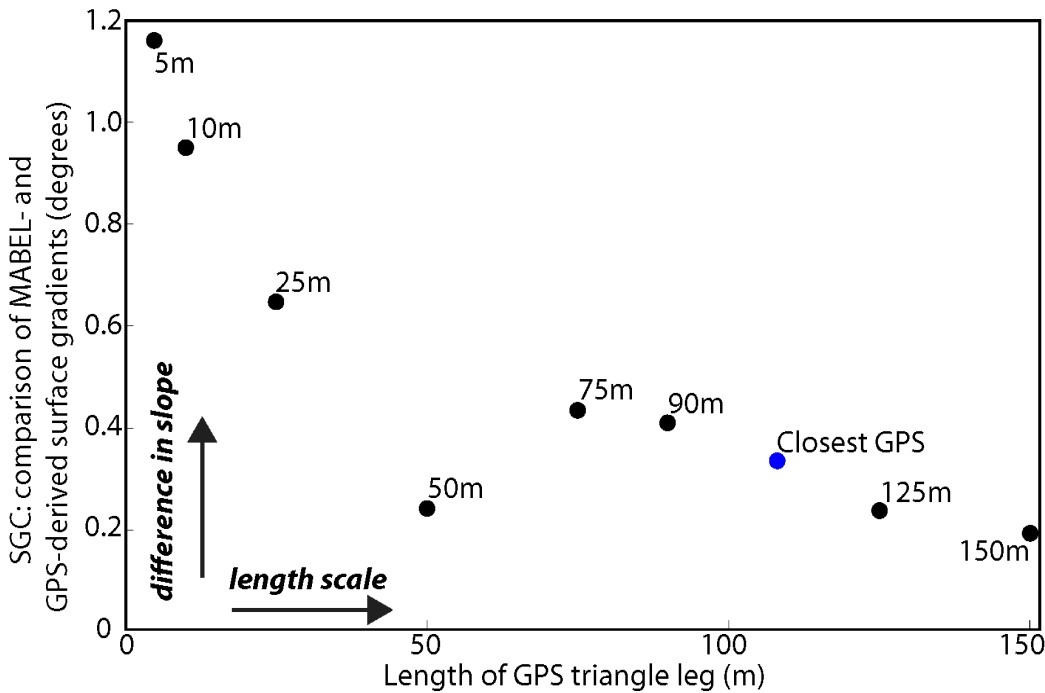

**Figure 9:** A surface-gradient comparison between a MABEL-derived surface (blue points in Fig. 3) and a series of GPS-derived surfaces, based on concentric equilateral triangles (black points here and in Fig. 3) and a surface based on the GPS survey sites that were closest to the nodes that defined the MABEL surface (blue point here and blue open circles in Fig. 3). The x-axis is the length of each side of the equilateral triangles (or a mean length, for the 'Closest GPS' surface); the y-axis is the surface-gradient comparison (SGC) parameter (defined in Eq. 1), or the RSS of the difference in surface gradient ($\delta z/\delta x$ and $\delta z/\delta y$), in degrees, between the MABEL-derived surface and each of the GPS-derived surfaces.

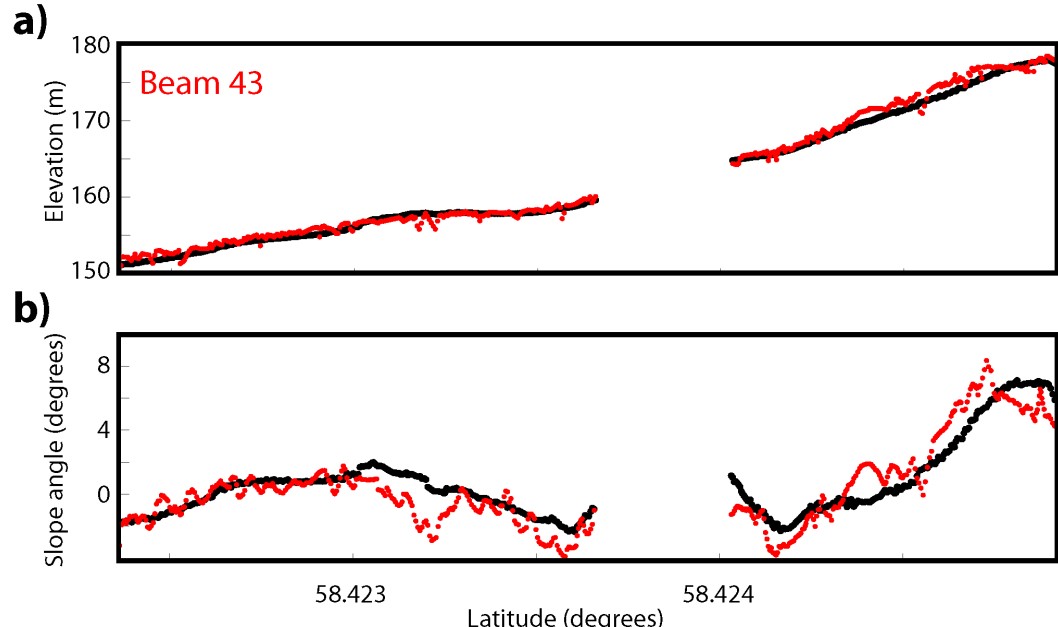

**Figure 10:** MABEL and DEM surfaces and slopes for a small stretch (see box in Fig. 8b) on the Lower Taku Glacier. **(a)** MABEL (red) and extracted DEM (black) elevations in m, for beam 43, migrated based on GPS data and corrected for an 8 m range bias. **(b)** MABEL (red) and DEM (black) across-track slope angle in degrees, using beams 43 and 50.