# Peer review of "MABEL photon-counting laser altimetry data in Alaska for ICESat-2"

_The Cryosphere, 2015_

## Referee Comment (RC1) · Anonymous Referee #1 · 2 May 2016

Review of submitted manuscript:
"MABEL photon-counting laser altimetry data in Alaska for ICESat-2 simulation sand development"
By K. Brunt, et al.

General Comments:

This paper presents an evaluation of MABEL data over selected regions with the motivation of proving the performance of a single-photon counting lidar system in support of the upcoming ICESat-2 mission. In particular, the authors are investigating if the planned collection strategy for ICESat-2 will support accurate assessment of local slope in order to determine the difference between surface slope and true elevation change over the ice sheets. The study is also intended to determine the resolution of the measurements in terms of identifying small melt ponds and crevasses within the surveyed area. The paper is written well and provides a thorough analysis of the data collected by MABEL. However, the true connection of MABEL with ATLAS, the ICESat-2 onboard instrument, isn't quite clearly presented. As such, the manuscript provides knowledge about MABEL and allows for a generalized confidence in the ICESat-2 mission but certainly could expand on comparison of the two systems particularly with the radiometric differences, variations in processing schemes and length scales beyond 70 cm sampling rates. It would also be beneficial to specifically relate the MABEL statistics with the quantitative performance goals (science requirements) of the satellite with respect to ice sheet elevation change derivation. That said, this is still an important piece of work as the community looks toward another space-based laser altimetry mission and it is definitely relevant to this journal.

Specific comments.

- In the abstract the purpose of MABEL is to support geophysical algorithm development, simulate key elements of the sampling strategy and assess elements of the resulting data that may vary seasonally. The last programmatic goal of MABEL seems out of place in terms of truly providing relevant information in terms of trying to separate out the errors in the MABEL measurements from seasonal variations. Later in the Introduction this goal isn't mentioned and other goals are added in; the text isn't consistent.
- This publication submission and the one from the author in 2014 (similar topic) cite the same reference with respect to the details of ICESat-2 but the specifications are not consistent (e.g. 10 m footprint versus 14 m footprint). Assumedly the specifications of the instrument have changed over the last 6 years since the reference was published? What else has changed? Isn't there another citation more current?
- Shouldn't there be some discussion on how the MABEL data compares to ATLAS beyond just data density (e.g. photons interpreted as signal for a given length scale)? Does the density impact the performance statistics of slope determination? It seems that data gaps hinder

characterization of the surface (elevation change), as would environmental impacts associated with topography and radiometry/reflectance.

- The value of MABEL is undeniable as a test bed instrument for ATLAS but it isn't clear in this publication how it directly simulates ICESat-2 performance in a quantitative way. Many of the conclusions are vague analysis language such as " the analysis proves that ICESat-2 will be able to provide a robust assessment of across-track slope". What does robust mean in this context? Valid? Precise? Consistent performance?
- Why isn't the m-atlas data used in this situation?
- Will the same surface interpretation algorithm be used on ICESat-2 data as MABEL? Does the processing scheme implemented have any impact on the comparable performance? How does the surface interpretation algorithm change with changing data density (i.e. laser power degradation over time)?
- If MABEL has a horizontal error of 2 m, how does this impact the slope comparison with the GPS surveys?
- What is the gridding resolution used on the MABEL data for in situ slope comparison? Does this cause any aliasing?
- Were there any other conclusions to the goal of is there sub-surface sampling happening other than '532 nm appears to be sub-surface sampling but there is also an after pulse". It was hard to determine the results of this investigation thread.
- MABEL beams are said to have non-uniform transmit energy? Is this average pulse energy relative to other beams or are there spatial differences of the energy distribution (Gaussian distribution ), or both? The comment "generally not the same pulse shape" is ambiguous. Does the changing or different pulse shapes have an impact on this study?
- Are the unique beam range biases on MABEL only due to the optical path of each beam? It seems like there are other influences on the ranges than just optical path but the text implies this is the only reason.
- The sentence on page 6, lines 1-2 doesn't seem to make sense or is incomplete.
- The authors rely quite a bit on the along-track signal density. As such, it might be helpful to present a table with MABEL performance (density statistics) comparison to what is expected with ICESat-2 design cases under certain conditions (radiometric conditions, topography, weather). This could be presented as an augmented Table 1.
- Does the noise in the process (background and detector noise) affect the interpretation of the surface and the subsequent determination of local slope?
- Does this MABEL analysis provide confidence that ICESat-2 will satisfy its science requirements? There is extensive elevation bias and precision discussion for MABEL for this study area but none of the results are projected to a quantitative ICESat-2 performance. Is that projection relevant here? Will a user get similar precision from ICESat-2 measurements for a single pass over this area and see the same detail of the surface topography?
- Page 11, line 27 seems to indicate that the 1064 nm beam would penetrate the water surface, which is not the case.

- Can you address the relationship to length scale along-track and the derived performance associated with accuracy and precision of the MABEL measurements? How do the length scales translate to ICESat-2?
- How do your conclusions of MABEL performance metrics allow for accurate change detection as related to ICESat-2 expected performance?

---

## Referee Comment (RC2) · Anonymous Referee #2 · 11 May 2016

Using photon-counting lidar is a new method for mapping ice sheets and glaciers. The paper presents the first results obtained by the MABEL system over complex glacier surfaces, such as heavily crevassed glaciers and lakes during melt conditions. These results are vital for assessing the performance of the ATLAS system to be flown on ICESat-2. They also highlight some of the advantages of the dense spatial sampling of photon-counting laser altimetry, for example for estimating crevasse geometry or melt pond shape and depth.

The paper could benefit from reorganizing the results and discussion sections by organizing them according to the surface feature types and measurement goals, instead of cycling through the different sites and goals several times. For example, the description of the altimetry profiles, photon histograms, and drop in the number of background photons over melt ponds are presented in three different sections. Also, some of the

figures are too minuscule to appreciate the details of the surface as depicted by the photon-counting system. For example, parts of the Bagley Icefield MABEL transect and corresponding photographs (Figure 6) covering the melt ponds and crevasses could be enlarged to the surface features (photograph) and structure (photons) motor clearly. The drop in the number of background photons over the melt ponds is very evident in Fig. 6.b – a beautiful example. However, the importance of this observation is almost lost in the details. Less convincing is the explanation about the melt pond depth determination. Showing the major elements of the melt pond in Figure 7 would help. Where is the elevation corresponding to the surface of the melt pond? And the bottom? Comparison of Figures 3 and 7 suggest that either the surface of the melt ponds is rough, or the maximum return is from below the surface. Can the authors distinguish between these cases? Also, I assume that the $\sim$0-2 photons per bin between 1397.5 and 1399.8 meters is due to returns from the ponds, i.e., volume scattering. Is this correct? Does the histogram depict the bottom of the pond?

The connection between the results obtained by the MABEL measurements and those expected from ATLAS is not well articulated. In particular, the paper should explain better how slope accuracy and spatial scales impact the accuracy of change detection with ATLAS. Also, the errors of the MABEL measurements should be better quantified to assess its performance and the implications for ATLAS/ICESat-2. For example, on page 4, line 29- page 5, line 3, the authors mention a different range bias of each of the MABEL beams. As the local slope is determined from elevations measured by different beams, this bias is expected to have an impact on the accuracy of slope determination. How stable are the range biases? How were the beams calibrated to one another? Was there any pointing bias, which would translate to additional elevation and slope errors? Page 7, lines 7-9 describe a calibration over ocean surface but does not mention how stable the offset was and if there was a pointing bias or not.

The explanation of DEM "migration" over the Lower Taku Glacier needs an overhaul. It is very hard to follow the details. Maybe a map with velocity vectors could help? What

is the expected accuracy of the WV-2 DEM? Page 13, line 1-6: what is "the difference between the DEM and the true elevation"?. Lines 4-6: was the MABEL range bias determined from ocean measurements? How was the surface melt estimated, any reference?

Detailed comments:

Mention the wavelength domains for 532 nm – green, and 1064 nm – near infrared and briefly summarize the current knowledge of penetration, surface and volume scattering with references.

Page 2, line 29: "ATLAS model"? Do you refer to the sensor model of ATLAS?

Page 2-3: A figure comparing ATLAS and MABEL geometries would be helpful

Page 3, 9-12: A brief summary about the expected accuracy and potential interpretation difficulties of the photon-counting altimetry in winter and summer conditions would be useful, e.g., penetration depth of green laser beam or impact of high surface reflectance on range bias (dead time issue)

Page 4, line 3-5: What assumptions are made to classify the photons into signal and noise/background?

Page 5, line 13: what is the resolution of the camera? Number of pixels, rows/columns? Is it a B/W or color camera? How often were images taken? Was there an overlap between consecutive images at the nominal flying height?

Page 5, line 25: Was the "standard" Landsat 8 spectral reflectance product used or did the authors derive their own reflectance? What wavelengths were used for the model? Was the panchromatic band used for Bagley Field because of the better spatial resolution?

Page 8, line 17: how large was the slope caused by wind stress or dynamic ocean topography? How large is the geoid undulation?

Page 8, line 27: under some operational conditions, such as??

Page 9, line 1: Was the second pulse removed by visual inspection and manual editing?

Page 10, line 20: flat, HORIZONTAL surface?

---

## Author Comment (AC2) · 17 Jun 2016

**Response to Anonymous Referee #2:**

We thank the anonymous reviewer for their very detailed and highly constructive comments. Edits based on your input (and that of the other reviewer) have substantially improved this manuscript.

Here are our responses (in red) to your specific comments (in black):

Using photon-counting lidar is a new method for mapping ice sheets and glaciers. The paper presents the first results obtained by the MABEL system over complex glacier surfaces, such as heavily crevassed glaciers and lakes during melt conditions. These results are vital for assessing the performance of the ATLAS system to be flown on ICESat-2. They also highlight some of the advantages of the dense spatial sampling of photon-counting laser altimetry, for example for estimating crevasse geometry or melt pond shape and depth.

The paper could benefit from reorganizing the results and discussion sections by organizing them according to the surface feature types and measurement goals, instead of cycling through the different sites and goals several times. For example, the description of the altimetry profiles, photon histograms, and drop in the number of background photons over melt ponds are presented in three different sections.

In reading through the manuscript and incorporating the edits based on review, we have reorganized the manuscript a bit, but kept it categorized in a series of topics (photon density, bias/precision, surface characterization, and slope). While we realize some things will still be repeated in this simplified structure, we have worked to minimize that repetition. Other structural changes include: 1) we moved some MABEL beam detail out of the Intro and combined it with other beam detail in the Data section; 2) we merged the Lower Taku WorldView and GPS data sections; 3) we merged the 'Slope' results, independent of flat or steep terrain; and, most importantly, 4) we have shortened and reworked the Lower Taku Results section.

Also, some of the figures are too minuscule to appreciate the details of the surface as depicted by the photon-counting system. For example, parts of the Bagley Icefield MABEL transect and corresponding photographs (Figure 6) covering the melt ponds and crevasses could be enlarged to the surface features (photograph) and structure (photons) motor clearly. The drop in the number of background photons over the melt ponds is very evident in Fig. 6.b – a beautiful example. However, the importance of this observation is almost lost in the details.

We agree. We note that many of the figures have new numbers, based on the reorganization of the manuscript. Most of the figures were simply made larger (*e.g.,* Figs. 2, 3, 4, 7, 9, and 10). But 3 others had to be reworked such that their details became more apparent (*e.g.,* Figs. 5, 6, and 8). And we added a new figure to address one of the reviewer's detailed comments. We believe we have addressed the reviewers concerns.

Less convincing is the explanation about the melt pond depth determination. Showing the major elements of the melt pond in Figure 7 would help. Where is the elevation

corresponding to the surface of the melt pond? And the bottom? Comparison of Figures 3 and 7 suggest that either the surface of the melt ponds is rough, or the maximum return is from below the surface. Can the authors distinguish between these cases? Also, I assume that the ~0-2 photons per bin between 1397.5 and 1399.8 meters is due to returns from the ponds, i.e., volume scattering. Is this correct? Does the histogram depict the bottom of the pond?

The authors agree that the analysis of the photon data over the melt ponds allows for debate. We have added elements to the figure, including surface and the after-pulse. We agree that there isn't a distinct feature on the histograms that suggests a return from the bottom of the pond; we have added this text to the manuscript. We also assume that the spread of the 532-nm histogram is associated with volume scattering throughout the pond; we have added that language as well. Some of the differences between the 2 histogram figures (formerly 3 and 7) are the result of variable length scales and, thus, very different photon counts associated with each analysis.

The connection between the results obtained by the MABEL measurements and those expected from ATLAS is not well articulated. In particular, the paper should explain better how slope accuracy and spatial scales impact the accuracy of change detection with ATLAS. Also, the errors of the MABEL measurements should be better quantified to assess its performance and the implications for ATLAS/ICESat-2.

We received two excellent reviews of this manuscript. Both Reviewer 1 and 2 make note of the connection between MABEL and ATLAS. We have added a bit of text throughout to address this main criticism (we have captured much of this in our responses to specific comments below). Most notably, there is a large paragraph at the start of the Discussion section that addresses this in detail. In summary, given that the measured MABEL signal-photon density is generally less than that predicted for ATLAS, and given our lack of quantitative knowledge of the throughput of MABEL, it is hard for us to scale MABEL results to ATLAS. However, what we can say is that *"if the ATLAS signal-photon density and signal-to-noise ratios are within 30% of its measurement requirements (and thus mimics the MABEL performance documented in this study), ATLAS can be used to measure surface slopes over both relatively flat ice-sheet interior conditions and steeper glaciers such as the Lower Taku Glacier, and identify melt ponds."* However, we cannot state further conclusions (e.g., accuracy and spatial scale comparisons), as these would require the instruments to have more similar radiometry.

For example, on page 4, line 29- page 5, line 3, the authors mention a different range bias of each of the MABEL beams. As the local slope is determined from elevations measured by different beams, this bias is expected to have an impact on the accuracy of slope determination. How stable are the range biases? How were the beams calibrated to one another? Was there any pointing bias, which would translate to additional elevation and slope errors? Page 7, lines 7-9 describe a calibration over ocean surface but does not mention how stable the offset was and if there was a pointing bias or not.

The reviewer is correct: MABEL's overall geolocation, and thus range issues, are an area of ongoing engineering discussions and limit this lidar from being utilized for a broader range of altimetry applications. However, the range biases are suitable for the overall goals of MABEL as they relate to ICESat-2 (algorithm development and error analysis).

As such, more rigorous assessments of bias and bias stability have not been part of prior MABEL deployment plans (*e.g.,* targeting many calibration targets throughout an individual flight). The previous deployments have included 'pitch-and-roll' maneuvers over open ocean (which supports a calibration that should minimize pointing biases) and flights over stable surfaces that have been surveyed (*e.g.,* Brunt et al., 2014 describes results from flying over well-surveyed airport departure aprons); language that addresses this has been added to the text:

*"Prior to Level 2A data processing, MABEL ranges are corrected for these channel-specific optical path lengths using a calibration derived from data recorded during aircraft pitch and roll maneuvers performed over stretches of open ocean. We assume that this calibration mitigates the larger channel biases, including those associated with errors in pointing. However, other smaller-scale channel biases may still exist; these smaller-scale channel bias corrections were on the order of decimeters."*

But for some of the slope assessments done here, we had to make assumptions on bias and bias stability. In this manuscript, we describe, and have expanded upon, our calibration technique in section 3.1, where we present the histograms over the open ocean. We choose a calibration beam and then subtract the mean differences between the other beams and the calibration beam to calibrate them to one another. For the histograms, we also detrend the data to remove long-wavelength ocean effects. We have modified the text to include these details; and we added detrending summary text:

*"The detrending of each beam takes into account all of these effects; this correction ranged from 0.11 to 0.29 m over the 3000 m of along-track data used for this analysis."*

For the slope assessments made on the Juneau Icefield, we make the assumption that the calibration is valid over the 75 km between the ocean site and the survey site (we have added this caveat to the language).

The explanation of DEM "migration" over the Lower Taku Glacier needs an overhaul. It is very hard to follow the details. Maybe a map with velocity vectors could help? What is the expected accuracy of the WV-2 DEM? Page 13, line 1-6: what is "the difference between the DEM and the true elevation"?. Lines 4-6: was the MABEL range bias determined from ocean measurements? How was the surface melt estimated, any reference?

We agree. We have substantially shortened and reworked the Lower Taku Results section. We believe this makes this section read much better. And the reviewer makes an excellent suggestion: we have added the velocities (and scaled vectors) to the Figure. The expected elevation accuracy of the WV DEM is on the order of meters (which is described in the section mentioned by the reviewer; Page 13, line 1-6; we have added a new reference, Shean et al., 2016, for this as well); this is effectively the difference between the *WorldView-2* DEM *elevation* and the true elevation. MABEL range bias was determined using the open ocean data in Figure 4. All of these details have been added to the text of this section. The surface melt was assessed at the GPS sites to be 2.3 m using ablation wires, by one of the co-authors; we have added this text to the manuscript.

Detailed comments:

Mention the wavelength domains for 532 nm – green, and 1064 nm – near infrared and briefly summarize the current knowledge of penetration, surface and volume scattering with references.

This is an excellent addition to the manuscript. We edited existing text and added the following:

*"MABEL (discussed in detail in McGill et al., 2013) is a multibeam, photon-counting lidar, sampling at both 532 (green) and 1064 (near infrared) nm wavelengths using short (~1.5 ns) laser pulses. The dual wavelength instrument design was intended to assess green-wavelength light penetration in water or snow (McGill et al., 2013). Deems et al. (2013) provides a review of lidar use for snow studies and describes how light at 532 and 1064 nm wavelengths interacts with snow surfaces. Light penetration into a snow surface is a function of both grain size (with larger snow-grain size resulting in increased volumetric scattering, and therefore increased light penetration) and wavelength (with 532 nm light having lower absorption than 1064 nm light, which ultimately produces increased light penetration at the shorter wavelength). Deems et al. (2013) also note that light penetration into snow surfaces is extremely difficult to accurately measure."*

Page 2, line 29: "ATLAS model"? Do you refer to the sensor model of ATLAS?

We were referring to the ATLAS instrument performance model. However, in light of other edits associated with the MABEL/ATLAS comparisons, this has been deleted.

Page 2-3: A figure comparing ATLAS and MABEL geometries would be helpful

Agreed. We adapted a figure from Brunt et al. (2014) to only include ICESat-2 and MABEL. Additions to the figure include across-track sampling length-scales.

Page 3, 9-12: A brief summary about the expected accuracy and potential interpretation difficulties of the photon-counting altimetry in winter and summer conditions would be useful, e.g., penetration depth of green laser beam or impact of high surface reflectance on range bias (dead time issue)

This is a great comment and makes for a great addition to the manuscript. We have added language that directly addresses this comment. The text includes language associated with NIR and green studies, which are still ongoing:

*"In winter, increased albedo, reduced ice-sheet surface roughness, and reduced solar background and backscatter in the atmosphere all lead to an increased signal-to-noise ratio and an increase in photon-retrieval density (i.e., the number of, and temporal distribution of photons transmitted and recorded by the lidar). In general, with increased photon-retrieval density, we expect better surface measurement precision. In the extreme case, the photon-retrieval density may be sufficiently high that the instrument receiver does not have the time required to process the incoming photon information before receiving more. This effect is referred to as 'instrument dead time' and can produce a positive surface elevation bias. In summer, reduced albedo, increased ice-sheet surface roughness, and increased solar background leads to a decrease in photon-retrieval density and signal-to-noise ratios, compromising measurement precision. The Alaska 2014 campaign also aimed to investigate how light at 532 and 1064 nm wavelengths interacts with the surface in melting conditions, and how this may affect the statistics of the 532 nm signal photons and overall elevation accuracy."*

Page 4, line 3-5: What assumptions are made to classify the photons into signal and noise/background?
While the details are discussed in Brunt et al. (2014), we agree with the reviewer that his section needed a few more details, especially with respect to the assumptions. We have added the text:
*"The algorithm is based on histograms of photon arrival times in 25 m along-track segments and 10 m vertical bins and assumes a random distribution of background photons and a symmetric return pulse. Further details of this surface-finding algorithm are described in Brunt et al. (2014). The GSFC algorithm is applicable to a wide range of surface types, while most ICESat-2 standard data product algorithms are surface-type specific (e.g., glacier, sea ice, ocean, vegetation, etc.) and more rigorous with respect to returns identified as surface signal."*

Page 5, line 13: what is the resolution of the camera? Number of pixels, rows/columns? Is it a B/W or color camera? How often were images taken? Was there an overlap between consecutive images at the nominal flying height?
Again, we agree with the reviewer that more detail is needed. We have added the following detail about the camera: 1) 6000x4000 pixels per image; 2) at 65,000 feet, this is approximately <3 m/pixel; 3) color images; and 4) 3-second intervals for ~30% overlap.

Page 5, line 25: Was the "standard" Landsat 8 spectral reflectance product used or did the authors derive their own reflectance? What wavelengths were used for the model? Was the panchromatic band used for Bagley Field because of the better spatial resolution?
We edited and added the following text to the Landsat discussion:
*"We assessed the performance of OLI's coastal blue, blue, green, red, and panchromatic channels in retrieving supraglacial lake depths. Ultimately, the models establish a relationship between Landsat 8 top-of-atmosphere (TOA) comparing pre-drainage spectral reflectance values over the lakes with a post-drainage digital elevation model (DEM), derived from WorldView-2 imagery acquired from the Polar Geospatial Center at the University of Minnesota, using image-processing software (ERDAS). Our analysis indicated that for shallow lakes (depth < 5 m), red and panchromatic band data are most suitable for supraglacial bathymetry. Because of the relatively small size of the lakes in our study area, we chose the panchromatic channel for the better spatial resolution."*

Page 8, line 17: how large was the slope caused by wind stress or dynamic ocean topography? How large is the geoid undulation?
The reviewer brings up an excellent point. Our goal was to use a stretch of open ocean to calibrate the beam elevations to one another in a relative sense. We used a 3-km stretch of data to do this. This should be significantly smaller than the length-scale of variations in dynamic topography and geoid undulation (changing ~1 to 10 m over length scales of ~100 to 1000 km). Wind stress is probably a bigger term on our 3-km length scale. While we can't separate these terms, detrending the data should account for all of these terms simultaneously. We have therefore provided the numbers associated with the detrending:

*"We calibrated the beam elevations to one another to remove the unique beam elevation biases; relative bias corrections ranged from 0.03 to 0.73 m. We then detrended the surface elevations based on a linear fit to the signal photons to remove any elevation differences associated with wind stress or the relatively small effects of ocean dynamic topography and geoid undulation. The detrending of each beam takes into account all of these effects; this correction ranged from 0.11 to 0.29 m over the 3000 m of along-track data used for this analysis."*

Page 8, line 27: under some operational conditions, such as??
The causes of the after-pulse are still not fully understood. We added the following language that explains this and provides more constraints on what we do know, especially with respect to the instrument/aircraft configuration of MABEL:
*"The exact conditions for after-pulsing are not completely understood, but are most likely the result of temperature drifts in the fundamental laser system. These occur due to changing environmental conditions within the instrument pod in the aircraft, and/or changes in efficiency of the coolant system. The cooling system relies upon passive external fins exposed to ambient cold conditions at altitude and these conditions (temperature, airflow) change during flight. The secondary laser pulses are primarily seen in the 1064 nm returns, and are minimized when the 1064 nm source is frequency-doubled to generate 532 nm beams."*

Page 9, line 1: Was the second pulse removed by visual inspection and manual editing?
We manually removed these when doing statistical analysis. We have added this to the text.

Page 10, line 20: flat, HORIZONTAL surface?
Yes, that is correct. However, based on other edits, and for continuity in this section, we have replace 'flat, HORIZONTAL' surface with *'detrended'* surface.

---

## Author Response (AR1)

**Response to Anonymous Referee #1:**

We thank the anonymous reviewer for their very detailed and highly constructive comments. Edits based on your input (and that of the other reviewer) have substantially improved this manuscript.

Here are our responses (in red) to your specific comments (in black):

This paper presents an evaluation of MABEL data over selected regions with the motivation of proving the performance of a single-photon counting lidar system in support of the upcoming ICESat-2 mission. In particular, the authors are investigating if the planned collection strategy for ICESat-2 will support accurate assessment of local slope in order to determine the difference between surface slope and true elevation change over the ice sheets. The study is also intended to determine the resolution of the measurements in terms of identifying small melt ponds and crevasses within the surveyed area. The paper is written well and provides a thorough analysis of the data collected by MABEL. However, the true connection of MABEL with ATLAS, the ICESat-2 onboard instrument, isn't quite clearly presented. As such, the manuscript provides knowledge about MABEL and allows for a generalized confidence in the ICESat-2 mission but certainly could expand on comparison of the two systems particularly with the radiometric differences, variations in processing schemes and length scales beyond 70 cm sampling rates. It would also be beneficial to specifically relate the MABEL statistics with the quantitative performance goals (science requirements) of the satellite with respect to ice sheet elevation change derivation. That said, this is still an important piece of work as the community looks toward another space-based laser altimetry mission and it is definitely relevant to this journal.

We received two excellent reviews of this manuscript. Both Reviewer 1 and 2 make note of the connection between MABEL and ATLAS. We have added a bit of text throughout to address this main criticism (we have captured much of this in our responses to specific comments below). Most notably, there is a large paragraph at the start of the Discussion section that addresses this in detail. In summary, given that the measured MABEL signal-photon density is generally less than that predicted for ATLAS, and given our lack of quantitative knowledge of the throughput of MABEL, it is hard for us to scale MABEL results to ATLAS. However, what we can say is that *"if the ATLAS signal-photon density and signal-to-noise ratios are within 30% of its measurement requirements (and thus mimics the MABEL performance documented in this study), ATLAS can be used to measure surface slopes over both relatively flat ice-sheet interior conditions and steeper glaciers such as the Lower Taku Glacier, and identify melt ponds." Please see our other comments below that directly address the ATLAS/MABEL concerns.*

Specific comments.

 $\cdot$  In the abstract the purpose of MABEL is to support geophysical algorithm development, simulate key elements of the sampling strategy and assess elements of the resulting data that may vary seasonally. The last programmatic goal of MABEL seems out of place in terms of truly providing relevant information in terms of trying to separate out the errors

in the MABEL measurements from seasonal variations. Later in the Introduction this goal isn't mentioned and other goals are added in; the text isn't consistent.

The reviewer makes a great point. The 'seasonal' language stems from a comparison between the timing of this AK campaign (July/August) and an earlier campaign based in Iceland (April/May). We have changed the text to separate overall MABEL goals with the goals of this particular deployment (which makes the abstract stronger). Further, we have made the language between the Abstract and the third and fourth paragraphs of the Introduction consistent. The Abstract (which summarizes the overall change) now reads:

"Given the new technology of ATLAS, an airborne instrument, the Multiple Altimeter Beam Experimental Lidar (MABEL), was developed to provide data needed for satellitealgorithm development and ICESat-2 error analysis. MABEL was deployed out of Fairbanks, Alaska in July 2014 to provide a test data set for algorithm development in summer conditions with water saturated snow and ice surfaces."

• This publication submission and the one from the author in 2014 (similar topic) cite the same reference with respect to the details of ICESat-2 but the specifications are not consistent (e.g. 10 m footprint versus 14 m footprint). Assumedly the specifications of the instrument have changed over the last 6 years since the reference was published? What else has changed? Isn't there another citation more current?

The reviewer is correct. Some aspects of ICESat-2 have changed since the Abdalati et al. (2010) reference. We have noted this in the first paragraph of the Introduction. An update (Markus et al., *submitted*) has recently been submitted to Remote Sensing of the Environment. We have added that reference. Text changes include:

"Abdalati et al. (2010) provides an early overview of the ATLAS concept and overall design. While the measurement goals of ATLAS remain as described in Abdalati et al. (2010), some of the details have evolved (Markus et al., submitted)."

 $\cdot$  Shouldn't there be some discussion on how the MABEL data compares to ATLAS beyond just data density (e.g. photons interpreted as signal for a given length scale)? Does the density impact the performance statistics of slope determination? It seems that data gaps hinder characterization of the surface (elevation change), as would environmental impacts associated with topography and radiometry/reflectance.

Density and data gaps definitely have an impact on the characterization of surface slope; Figs. 8 and 10 indicate how clouds lead to gaps in the assessment. We have added a fair bit of text to the MABEL instrument and data descriptions that address instrumentation differences. Further, we have added a paragraph at the start of the Discussion section that addresses MABEL/ATLAS comparison issues, and thus directly addresses some of this comment. However, since MABEL measured signal-photon densities were not as high as the predicted ATLAS values, and given our lack of quantitative knowledge of the throughput of MABEL, it was hard for us to scale MABEL results to ATLAS and provide quantitative ATLAS performance statistics.

• The value of MABEL is undeniable as a test bed instrument for ATLAS but it isn't clear in this publication how it directly simulates ICESat-2 performance in a quantitative way. Many of the conclusions are vague analysis language such as " the analysis proves

that ICESat-2 will be able to provide a robust assessment of across-track slope". What does robust mean in this context? Valid? Precise? Consistent performance?

This is similar to the comment above, and to the overall critique from this reviewer. In the new Discussion paragraph, we acknowledge that due to the fact that MABEL measured signal-photon densities were not as high as the predicted ATLAS values, and given our lack of quantitative knowledge of the throughput of MABEL, it was hard for us to scale MABEL results to ATLAS, and thus make direct quantitative assessments. However, we agree with the reviewer that 'robust' was not a meaningful word; we have replaced it with 'valid' in one instance and 'non-aliased' in another.

**• Why isn't the m-atlas data used in this situation?**

M-ATLAS is a dataset produced by the ICESat-2 Project Science Office. It is primarily intended for ATLAS algorithm testing. This dataset uses MABEL data to simulate expected ATLAS photon point clouds to more closely match our model predictions of ATLAS instrument performance. M-ATLAS is intended to assess photon point clouds over homogeneous surfaces, where combined beams are looking at a similar surface. Much of the analysis in this study focuses on heterogeneous surfaces, which include small-scale features that are not common from beam to beam (small melt ponds and crevasses). Thus, we did not look at the M-ATLAS data products. Since this is a very preliminary dataset, used primarily for internal purposes, and since readers will not be familiar of this dataset, we prefer to refrain from incorporating it into this manuscript.

 $\cdot$  Will the same surface interpretation algorithm be used on ICESat-2 data as MABEL? Does the processing scheme implemented have any impact on the comparable performance? How does the surface interpretation algorithm change with changing data density (i.e. laser power degradation over time)?

The NASA GSFC surface finder is a generic surface-finding algorithm used for a wide range of surface types, and therefore a conservative estimate of surface signal (it flags more photons as signal); ATLAS algorithms will be specific to the surface type (*e.g.*, glacier, sea ice, ocean, vegetation, etc.) and more rigorous with respect to returns identified as surface signal (these will flag fewer photons as signal, relative to the GSFC algorithm). We have added language to address this. All of the algorithms have some form of along-track interpretation length scales. Thus, if data density changes, either as a result of environmental (*e.g.*, cloud cover) or instrumental (*e.g.*, degradation over time) effects, the surface finders are compromised. Since the along-track length-scales will be unique to the surface type, we prefer to leave a more formalized discussion of data density impact on algorithm surface detection for the ICESat-2 algorithm documents (in process).

 $\cdot$  If MABEL has a horizontal error of 2 m, how does this impact the slope comparison with the GPS surveys?

The reviewer makes a great point here. We believe that the averaging that is done to determine an elevation for MABEL at given points should sufficiently rectify the geolocation concern. The MABEL elevations that defined the MABEL surface for slope determination were based on the mean of signal photons around the points of intersection of the MABEL ground track and the trends of the GPS survey lines. However, based on

this comment, we slightly modified our approach. Originally, the MABEL mean elevations that defined the MABEL surface were based on the closest 100 signal photons to the point of intersection with the GPS trend; we have changed this to be signal photons within a 5-m radius, to address the reviewers concerns and to base the analysis on the length scale of geolocation error. This led to a slight change to Figure 5, but not in the overall conclusions. We added the following text to address the reviewer's concern:

"Elevations at those nodes were determined by taking an average of the elevations of the signal photons within a 5 m radius of those points, to take into account MABEL horizontal geolocation uncertainty."

 $\cdot$  What is the gridding resolution used on the MABEL data for in situ slope comparison? Does this cause any aliasing?

The MABEL grid is  $1 \ge 1 = m$ . This is sufficiently small for length scales appropriate to ICESat-2 slope determination (90 m). We have added this scale to the text.

 $\cdot$  Were there any other conclusions to the goal of is there sub-surface sampling happening other than '532 nm appears to be sub-surface sampling but there is also an after pulse". It was hard to determine the results of this investigation thread.

This is a great question and still an area of active research. Geolocation limitations compromised our ability to further interrogate this topic with this dataset. We have added the following text:

"Penetration of 532 nm wavelength light into the surface, be it a melt pond or snow, is an ongoing area of research for ICESat-2 algorithm development. MABEL geolocation uncertainty, and the fact that the 1064 and 532 nm beams do not have coincident footprints for more direct comparison (as the 1064 nm beams lead the 532 nm beams by ~60 m), compromised our ability to further interrogate this topic with this dataset, as the data could not be precisely co-registered spatially. Due to these limitations, a separate campaign with a different photon-counting laser altimeter (with both a more accurate geopositioning system and coincident 1064 and 532 nm footprints) was deployed to Thule, Greenland, in July and August 2015 (Brunt et al., 2015). Processing and analysis of that dataset are still ongoing.

Analysis of MABEL data over small melt ponds on the Bagley Icefield in Alaska provided a preliminary assessment of how green-wavelength photon-counting systems will interact with water on an ice surface..."

 $\cdot$  MABEL beams are said to have non-uniform transmit energy? Is this average pulse energy relative to other beams or are there spatial differences of the energy distribution (Gaussian distribution), or both? The comment "generally not the same pulse shape" is ambiguous. Does the changing or different pulse shapes have an impact on this study?

Our original statement was intended to relate the average pulse energy of the beams relative to each other. We have added this language to the text:

"Relative to one another, the MABEL beams have non-uniform average transmit energy."

But the reviewer made us also reconsider additional text in this paragraph. In reading the next couple of sentences, which contained the statement "transmit-pulse shapes are generally not the same", we realized that this was not our intent; the paragraph is really

discussing transmit energy for beams with unique optical paths. We have changed the statement to:

"...transmit-pulse energies are generally not equal."

We then removed the next sentence that continued the discussion about pulse shape. Based on previous MABEL results, pulse shapes have been fairly symmetric (from Brunt et al., 2014: "...analysis of MABEL data from the ice-sheet interior indicates that the returned pulse shape is largely symmetric."). Results here over open ocean (Figure 3) suggest the same. Changes to the pulse shape would undoubtedly have an effect on this. Broadening of the pulse would most likely lead to larger values for both accuracy and precision. Our estimates of accuracy and precision are consistent with previous results. Thus, we do not believe that MABEL pulse shapes are changing.

 $\cdot$  Are the unique beam range biases on MABEL only due to the optical path of each beam? It seems like there are other influences on the ranges than just optical path but the text implies this is the only reason.

There are certainly environmental conditions that could contribute to an overall bias (e.g., temperature of the laser). Our expectation is that environmental influences would have a uniform effect on all beams. We have added language that addresses this. However, the 'unique' nature of each beam bias stems from the unique optical path lengths (due to unique optical fiber lengths) of each beam through the instrument.

• The sentence on page 6, lines 1-2 doesn't seem to make sense or is incomplete. Agreed. We removed this sentence and added the necessary detail to other sentences in this paragraph.

• The authors rely quite a bit on the along-track signal density. As such, it might be helpful to present a table with MABEL performance (density statistics) comparison to what is expected with ICESat-2 design cases under certain conditions (radiometric conditions, topography, weather). This could be presented as an augmented Table 1. Great idea. We have added a row to Table 1 that provides ATLAS information, appropriate to the surface type and environmental conditions. For the open ocean, we have provided a range, as we do not know the wind state during MABEL data acquisition.

 $\cdot$  Does the noise in the process (background and detector noise) affect the interpretation of the surface and the subsequent determination of local slope?

We do not believe so. The MABEL data used for this particular analysis were identified as surface returns by the GSFC algorithm. While that algorithm is conservative and may misidentify background as signal, we believe that our averaging technique mitigates any adverse effect on the result. The MABEL elevations that defined the MABEL surface for slope determination were based on the mean of the signal photons within a 5-m radius of those points of intersection (generally 20 - 50 signal photons, which should be sufficient to average out the small number of background outliers).

 $\cdot$  Does this MABEL analysis provide confidence that ICESat-2 will satisfy its science requirements? There is extensive elevation bias and precision discussion for MABEL for

this study area but none of the results are projected to a quantitative ICESat-2 performance. Is that projection relevant here? Will a user get similar precision from ICESat-2 measurements for a single pass over this area and see the same detail of the surface topography?

This comment addresses the key critique identified by both reviewers of this manuscript. As we stated above, we have added a paragraph at the start of the Discussion section that addresses this comment. In that paragraph, we acknowledge that due to the fact that MABEL measured signal-photon densities were not as high as the predicted ATLAS values, and given our lack of quantitative knowledge of the throughput of MABEL, it was hard for us to scale MABEL results to ATLAS, and thus make direct quantitative assessments. However, what we can quantitatively say that:

"... if the ATLAS signal-photon density and signal-to-noise ratios are within 30% of its measurement requirements (and thus mimics the MABEL performance documented in this study), ATLAS can be used to measure surface slopes over both relatively flat ice-sheet interior conditions and steeper glaciers such as the Lower Taku Glacier, and identify melt ponds."

 $\cdot$  Page 11, line 27 seems to indicate that the 1064 nm beam would penetrate the water surface, which is not the case.

The reviewer is correct; this language was intended to compare how the different wavelengths interact with water, not to suggest that 1064 penetrates the surface. We have changed this language:

"This figure depicts how light at 532 and 1064 nm wavelengths interacts with the surface of the melt pond, and how the melt pond affects the statistics of the 532 nm return signal."

• Can you address the relationship to length scale along-track and the derived performance associated with accuracy and precision of the MABEL measurements? How do the length scales translate to ICESat-2?

For many of the performance metrics presented here, we used a 0.7 m along-track lengthscale for analysis, to match that of ICESat-2. Since absolute accuracy was not required for this analysis, we did not determine an overall bias for MABEL. For precision, we used a 3-km stretch of open water to assess the individual beams. Results here were comparable to those determined over a 1.4-km departure apron, in a previous study (Brunt et al., 2014). Thus, we are confident that these values represent overall MABEL surface measurement precision. However, as stated previously, we cannot directly, or quantitatively, scale MABEL performance or precision results to ATLAS.

 $\cdot$  How do your conclusions of MABEL performance metrics allow for accurate change detection as related to ICESat-2 expected performance?

This comment also addresses the reviewer's key critique of this manuscript. As we stated above, we have added a paragraph at the start of the Discussion section that addresses this comment. In that paragraph, we make only relative assessments of MABEL measured performance relative to ATLAS predicted performance. Ultimately, we summarize that if ATLAS performance is within 30% of its measurement requirements, ATLAS can be used to measure flat and steep surface slopes, and identify melt ponds. But we cannot

make further conclusions, as these would require the instruments to have more similar radiometry.

**Response to Anonymous Referee #2:**

We thank the anonymous reviewer for their very detailed and highly constructive comments. Edits based on your input (and that of the other reviewer) have substantially improved this manuscript.

Here are our responses (in red) to your specific comments (in black):

Using photon-counting lidar is a new method for mapping ice sheets and glaciers. The paper presents the first results obtained by the MABEL system over complex glacier surfaces, such as heavily crevassed glaciers and lakes during melt conditions. These results are vital for assessing the performance of the ATLAS system to be flown on ICESat-2. They also highlight some of the advantages of the dense spatial sampling of photon-counting laser altimetry, for example for estimating crevasse geometry or melt pond shape and depth.

The paper could benefit from reorganizing the results and discussion sections by organizing them according to the surface feature types and measurement goals, instead of cycling through the different sites and goals several times. For example, the description of the altimetry profiles, photon histograms, and drop in the number of background photons over melt ponds are presented in three different sections.

In reading through the manuscript and incorporating the edits based on review, we have reorganized the manuscript a bit, but kept it categorized in a series of topics (photon density, bias/precision, surface characterization, and slope). While we realize some things will still be repeated in this simplified structure, we have worked to minimize that repetition. Other structural changes include: 1) we moved some MABEL beam detail out of the Intro and combined it with other beam detail in the Data section; 2) we merged the Lower Taku WorldView and GPS data sections; 3) we merged the 'Slope' results, independent of flat or steep terrain; and, most importantly, 4) we have shortened and reworked the Lower Taku Results section.

Also, some of the figures are too minuscule to appreciate the details of the surface as depicted by the photon-counting system. For example, parts of the Bagley Icefield MABEL transect and corresponding photographs (Figure 6) covering the melt ponds and crevasses could be enlarged to the surface features (photograph) and structure (photons) motor clearly. The drop in the number of background photons over the melt ponds is very evident in Fig. 6.b – a beautiful example. However, the importance of this observation is almost lost in the details.

We agree. We note that many of the figures have new numbers, based on the reorganization of the manuscript. Most of the figures were simply made larger (*e.g.*, Figs. 2, 3, 4, 7, 9, and 10). But 3 others had to be reworked such that their details became more apparent (*e.g.*, Figs. 5, 6, and 8). And we added a new figure to address one of the reviewer's detailed comments. We believe we have addressed the reviewers concerns.

Less convincing is the explanation about the melt pond depth determination. Showing the major elements of the melt pond in Figure 7 would help. Where is the elevation

corresponding to the surface of the melt pond? And the bottom? Comparison of Figures 3 and 7 suggest that either the surface of the melt ponds is rough, or the maximum return is from below the surface. Can the authors distinguish between these cases? Also, I assume that the  $\sim$ 0-2 photons per bin between 1397.5 and 1399.8 meters is due to returns from the ponds, i.e., volume scattering. Is this correct? Does the histogram depict the bottom of the pond?

The authors agree that the analysis of the photon data over the melt ponds allows for debate. We have added elements to the figure, including surface and the after-pulse. We agree that there isn't a distinct feature on the histograms that suggests a return from the bottom of the pond; we have added this text to the manuscript. We also assume that the spread of the 532-nm histogram is associated with volume scattering throughout the pond; we have added that language as well. Some of the differences between the 2 histogram figures (formerly 3 and 7) are the result of variable length scales and, thus, very different photon counts associated with each analysis.

The connection between the results obtained by the MABEL measurements and those expected from ATLAS is not well articulated. In particular, the paper should explain better how slope accuracy and spatial scales impact the accuracy of change detection with ATLAS. Also, the errors of the MABEL measurements should be better quantified to assess its performance and the implications for ATLAS/ICESat-2.

We received two excellent reviews of this manuscript. Both Reviewer 1 and 2 make note of the connection between MABEL and ATLAS. We have added a bit of text throughout to address this main criticism (we have captured much of this in our responses to specific comments below). Most notably, there is a large paragraph at the start of the Discussion section that addresses this in detail. In summary, given that the measured MABEL signal-photon density is generally less than that predicted for ATLAS, and given our lack of quantitative knowledge of the throughput of MABEL, it is hard for us to scale MABEL results to ATLAS. However, what we can say is that *"if the ATLAS signal-photon density and signal-to-noise ratios are within 30% of its measurement requirements (and thus mimics the MABEL performance documented in this study), ATLAS can be used to measure surface slopes over both relatively flat ice-sheet interior conditions and steeper glaciers such as the Lower Taku Glacier, and identify melt ponds."* However, we cannot state further conclusions (e.g., accuracy and spatial scale comparisons), as these would require the instruments to have more similar radiometry.

For example, on page 4, line 29- page 5, line 3, the authors mention a different range bias of each of the MABEL beams. As the local slope is determined from elevations measured by different beams, this bias is expected to have an impact on the accuracy of slope determination. How stable are the range biases? How were the beams calibrated to one another? Was there any pointing bias, which would translate to additional elevation and slope errors? Page 7, lines 7-9 describe a calibration over ocean surface but does not mention how stable the offset was and if there was a pointing bias or not.

The reviewer is correct: MABEL's overall geolocation, and thus range issues, are an area of ongoing engineering discussions and limit this lidar from being utilized for a broader range of altimetry applications. However, the range biases are suitable for the overall goals of MABEL as they relate to ICESat-2 (algorithm development and error analysis).

As such, more rigorous assessments of bias and bias stability have not been part of prior MABEL deployment plans (*e.g.*, targeting many calibration targets throughout an individual flight). The previous deployments have included 'pitch-and-roll' maneuvers over open ocean (which supports a calibration that should minimize pointing biases) and flights over stable surfaces that have been surveyed (*e.g.*, Brunt et al., 2014 describes results from flying over well-surveyed airport departure aprons); language that addresses this has been added to the text:

"Prior to Level 2A data processing, MABEL ranges are corrected for these channelspecific optical path lengths using a calibration derived from data recorded during aircraft pitch and roll maneuvers performed over stretches of open ocean. We assume that this calibration mitigates the larger channel biases, including those associated with errors in pointing. However, other smaller-scale channel biases may still exist; these smaller-scale channel bias corrections were on the order of decimeters."

But for some of the slope assessments done here, we had to make assumptions on bias and bias stability. In this manuscript, we describe, and have expanded upon, our calibration technique in section 3.1, where we present the histograms over the open ocean. We choose a calibration beam and then subtract the mean differences between the other beams and the calibration beam to calibrate them to one another. For the histograms, we also detrend the data to remove long-wavelength ocean effects. We have modified the text to include these details; and we added detrending summary text:

"The detrending of each beam takes into account all of these effects; this correction ranged from 0.11 to 0.29 m over the 3000 m of along-track data used for this analysis."

For the slope assessments made on the Juneau Icefield, we make the assumption that the calibration is valid over the 75 km between the ocean site and the survey site (we have added this caveat to the language).

The explanation of DEM "migration" over the Lower Taku Glacier needs an overhaul. It is very hard to follow the details. Maybe a map with velocity vectors could help? What is the expected accuracy of the WV-2 DEM? Page 13, line 1-6: what is "the difference between the DEM and the true elevation"?. Lines 4-6: was the MABEL range bias determined from ocean measurements? How was the surface melt estimated, any reference?

We agree. We have substantially shortened and reworked the Lower Taku Results section. We believe this makes this section read much better. And the reviewer makes an excellent suggestion: we have added the velocities (and scaled vectors) to the Figure. The expected elevation accuracy of the WV DEM is on the order of meters (which is described in the section mentioned by the reviewer; Page 13, line 1-6; we have added a new reference, Shean et al., 2016, for this as well); this is effectively the difference between the *WorldView-2* DEM *elevation* and the true elevation. MABEL range bias was determined using the open ocean data in Figure 4. All of these details have been added to the text of this section. The surface melt was assessed at the GPS sites to be 2.3 m using ablation wires, by one of the co-authors; we have added this text to the manuscript.

Detailed comments:

Mention the wavelength domains for 532 nm – green, and 1064 nm – near infrared and briefly summarize the current knowledge of penetration, surface and volume scattering with references.

This is an excellent addition to the manuscript. We edited existing text and added the following:

"MABEL (discussed in detail in McGill et al., 2013) is a multibeam, photon-counting lidar, sampling at both 532 (green) and 1064 (near infrared) nm wavelengths using short (~1.5 ns) laser pulses. The dual wavelength instrument design was intended to assess green-wavelength light penetration in water or snow (McGill et al., 2013). Deems et al. (2013) provides a review of lidar use for snow studies and describes how light at 532 and 1064 nm wavelengths interacts with snow surfaces. Light penetration into a snow surface is a function of both grain size (with larger snow-grain size resulting in increased volumetric scattering, and therefore increased light penetration) and wavelength (with 532 nm light having lower absorption than 1064 nm light, which ultimately produces increased light penetration at the shorter wavelength). Deems et al. (2013) also note that light penetration into snow surfaces is extremely difficult to accurately measure."

Page 2, line 29: "ATLAS model"? Do you refer to the sensor model of ATLAS? We were referring to the ATLAS instrument performance model. However, in light of other edits associated with the MABEL/ATLAS comparisons, this has been deleted.

Page 2-3: A figure comparing ATLAS and MABEL geometries would be helpful Agreed. We adapted a figure from Brunt et al. (2014) to only include ICESat-2 and MABEL. Additions to the figure include across-track sampling length-scales.

Page 3, 9-12: A brief summary about the expected accuracy and potential interpretation difficulties of the photon-counting altimetry in winter and summer conditions would be useful, e.g., penetration depth of green laser beam or impact of high surface reflectance on range bias (dead time issue)

This is a great comment and makes for a great addition to the manuscript. We have added language that directly addresses this comment. The text includes language associated with NIR and green studies, which are still ongoing:

"In winter, increased albedo, reduced ice-sheet surface roughness, and reduced solar background and backscatter in the atmosphere all lead to an increased signal-to-noise ratio and an increase in photon-retrieval density (i.e., the number of, and temporal distribution of photons transmitted and recorded by the lidar). In general, with increased photon-retrieval density, we expect better surface measurement precision. In the extreme case, the photon-retrieval density may be sufficiently high that the instrument receiver does not have the time required to process the incoming photon information before receiving more. This effect is referred to as 'instrument dead time' and can produce a positive surface elevation bias. In summer, reduced albedo, increased ice-sheet surface roughness, and increased solar background leads to a decrease in photon-retrieval density and signal-to-noise ratios, compromising measurement precision. The Alaska 2014 campaign also aimed to investigate how light at 532 and 1064 nm wavelengths interacts with the surface in melting conditions, and how this may affect the statistics of the 532 nm signal photons and overall elevation accuracy." Page 4, line 3-5: What assumptions are made to classify the photons into signal and noise/background?

While the details are discussed in Brunt et al. (2014), we agree with the reviewer that his section needed a few more details, especially with respect to the assumptions. We have added the text:

"The algorithm is based on histograms of photon arrival times in 25 m along-track segments and 10 m vertical bins and assumes a random distribution of background photons and a symmetric return pulse. Further details of this surface-finding algorithm are described in Brunt et al. (2014). The GSFC algorithm is applicable to a wide range of surface types, while most ICESat-2 standard data product algorithms are surface-type specific (e.g., glacier, sea ice, ocean, vegetation, etc.) and more rigorous with respect to returns identified as surface signal."

Page 5, line 13: what is the resolution of the camera? Number of pixels, rows/columns? Is it a B/W or color camera? How often were images taken? Was there an overlap between consecutive images at the nominal flying height?

Again, we agree with the reviewer that more detail is needed. We have added the following detail about the camera: 1) 6000x4000 pixels per image; 2) at 65,000 feet, this is approximately
- 5 M. S. Moussavi5,6, K. M. Walsh2,7, W. B. Cook2, and T. Markus2
- 6 [1] {Earth System Science Interdisciplinary Center, University of Maryland, College
- 7 Park, Maryland}
- 8 [2] {NASA Goddard Space Flight Center, Greenbelt, Maryland}
- 9 [3] {University of Alaska Southeast, Juneau, Alaska}
- 10 [4] {University of Alberta, Edmonton, Alberta, Canada}
- 11 [5] {Cooperative Institute for Research in Environmental Sciences (CIRES), University
- 12 of Colorado, Boulder, Colorado}
- 13 [6] {National Snow and Ice Data Center (NSIDC), CIRES, University of Colorado,
- 14 Boulder, Colorado}
- 15 [7] {Stinger Ghaffarian Technologies, Inc., Greenbelt, Maryland}
- 16 Correspondence to: K.M. Brunt (kelly.m.brunt@nasa.gov)
- 17

**18 Abstract**

- 19 Ice, Cloud, and land Elevation Satellite-2 (ICESat-2) is scheduled to launch in late 2017
- 20 and will carry the Advanced Topographic Laser Altimeter System (ATLAS), which is a
- 21 photon-counting laser altimeter and represents a new approach to satellite determination
- 22 of surface elevation. Given the new technology of ATLAS, an airborne instrument, the
- 23 Multiple Altimeter Beam Experimental Lidar (MABEL), was developed to provide data
- 24 needed for satellite-algorithm development and ICESat-2 error analysis. MABEL was
- 25 deployed out of Fairbanks, Alaska in July 2014 to provide a test data set for algorithm
- 26 development in summer conditions with water saturated snow and ice surfaces. Here we
- 27 compare MABEL lidar data to in situ observations in Southeast Alaska to assess

**kelly brunt 5/17/2016 17:12**

**Deleted: ,**

kelly brunt 5/17/2016 17:13 Deleted: simulating key elements of the photon-counting sampling strategy, and assessing elements of the resulting data that may vary seasonally

kelly brunt 6/14/2016 13:25 Deleted:

1 instrument performance in summer conditions and in the presence of glacier surface melt 2 ponds and a wet snowpack. Results indicate that: 1) based on MABEL and in situ data 3 comparisons, the ATLAS 90 m beam-spacing strategy will provide a valid assessment of 4 across-track slope that is consistent with shallow slopes (<1°) of an ice-sheet interior over 5 50 to 150 m length scales; 2) the dense along-track sampling strategy of photon counting 6 systems can provide, crevasse detail; and 3) MABEL 532 nm wavelength light may 7 sample both the surface and subsurface of shallow (approximately 2 m deep) supraglacial 8 melt ponds. The data associated with crevasses and melt ponds indicate the potential 9 ICESat-2 will have for the study of mountain and other small glaciers.

10

**11 **1 Introduction**

12 Ice, Cloud, and land Elevation Satellite-2 (ICESat-2) is a NASA mission scheduled to 13 launch in 2017. ICESat-2 is a follow-on mission to ICESat (2003-2009) and will extend 14 the time series of elevation-change measurements aimed at estimating the contribution of polar ice sheets to eustatic sea level rise. ICESat-2 will carry the Advanced Topographic 15 16 Laser Altimeter System (ATLAS), which uses a different surface detection strategy than 17 the instrument onboard ICESat. Abdalati et al. (2010) provides an early overview of the 18 ATLAS concept and overall design. While the measurement goals of ATLAS remain as 19 described in Abdalati et al. (2010), some of the details have evolved (Markus et al., 20 submitted), ATLAS is a 6-beam, photon-counting laser altimeter (Fig. 1). In a photon-21 counting system, single-photon sensitive detectors are used, to record arrival time of any 22 detected photon, ATLAS will use short (< 2 ns) 532 nm (green) wavelength laser pulses, 23 with a 10 kHz repetition rate, which yields a ~0.70 m along-track sampling interval, and a 24  $\sim$ 17 m diameter footprint. An accurate assessment of ice-sheet surface-elevation change 25 based on altimetry is dependent upon knowledge of local slope (Zwally et al., 2011). 26 Therefore, the six ATLAS beams are arranged into three sets of pairs. Spacing between 27 the three pair sets is ~3 km to increase sampling density, while spacing between each 28 beam within a given pair will be ~90 m to make the critical determination of local slope 29 on each pass. Therefore, elevation change can be determined from only two passes of a 30 given area (Brunt et al., 2014).

**kelly brunt 5/26/2016 13:17 Deleted: robust**

| _ | kelly brunt 6/15/2016 12:00 |
|---|-----------------------------|
|   | Deleted: s                  |
|   | kelly brunt 6/15/2016 12:01 |
|   | Deleted: be                 |
|   | kelly brunt 6/15/2016 12:01 |
|   | Deleted: ing                |

| kelly brunt 5/17/2016 17:23                                                          |
|--------------------------------------------------------------------------------------|
| Deleted: Specifically,                                                               |
| kelly brunt 5/17/2016 17:23                                                          |
| Deleted: will be                                                                     |
| kelly brunt 6/14/2016 13:27                                                          |
| Deleted: , and the                                                                   |
| kelly brunt 6/14/2016 13:27                                                          |
| Deleted: is recorded                                                                 |
| kelly brunt 5/26/2016 14:37                                                          |
| Deleted: 14                                                                          |
| kelly brunt 6/14/2016 13:28                                                          |

[revised manuscript text omitted]
                                                                                                                                      |
| kelly brunt 6/14/2016 13:38                                                                                                                                                                                                                                    |
| Deleted: and then divided into a series of                                                                                                                                                                                                                     |
| kelly brunt 6/14/2016 13:39                                                                                                                                                                                                                                    |
| Deleted: beams                                                                                                                                                                                                                                                 |
| kelly brunt 5/25/2016 12:29                                                                                                                                                                                                                                    |
| Deleted: shapes                                                                                                                                                                                                                                                |
| kelly brunt 6/14/2016 13:39                                                                                                                                                                                                                                    |
| Deleted: the same                                                                                                                                                                                                                                              |
| kelly brunt 5/25/2016 12:34                                                                                                                                                                                                                                    |
| Deleted: Although MABEL does not digitize transmit pulse shapes, examining pulse shape differences over impenetrable targets ( e.g. , airport runways) can be considered a proxy when examining 1064 nm and 532 nm return pulse characteristics. |
| kelly brunt 6/14/2016 13:39                                                                                                                                                                                                                                    |
| Deleted: analysis                                                                                                                                                                                                                                              |
| kelly brunt 6/14/2016 13:39                                                                                                                                                                                                                                    |
| Deleted: analysis                                                                                                                                                                                                                                              |
| kelly brunt 6/14/2016 13:39                                                                                                                                                                                                                                    |
| Deleted: , in an along-track direction,                                                                                                                                                                                                                        |
| (kolly, brupt 6/11/2016 12:40                                                                                                                                                                                                                                  |
| Kelly brufil 6/14/2016 13.40                                                                                                                                                                                                                                   |
| Deleted: at                                                                                                                                                                                                                                                    |
| Deleted:         at           kelly brunt 6/14/2016 13:40                                                                                                                                                                                                      |

Because of the different optical paths each beam takes through the instrument, each 1 2 MABEL beam has a unique range bias (McGill et al., 2013). Prior to Level 2A data 3 processing, MABEL ranges are corrected for these channel-specific optical path lengths 4 using a calibration derived from data recorded during aircraft pitch and roll maneuvers performed over stretches of open ocean. We assume that this calibration mitigates the 5 larger channel biases, including those associated with errors in pointing. However, other 6 7 smaller-scale channel biases may still exist; these smaller-scale channel bias corrections were on the order of decimeters. Much of the analysis performed here, such as evaluation 8 9 of local surface slope, did not require absolute range accuracy. Therefore, the individual 10 beams were generally only calibrated with respect to one another based on data collected over the nearest flat surface (e.g., open water). These calibrations were made relative to 11 12 the beam closest to the center of the array.

13

**14 2.2 MABEL camera imagery**

15 For the 2014 Alaska campaign, a camera was integrated with MABEL and was 16 successful for over 40% of the campaign's duration. The images were typically used to 17 visually confirm the type of surface being overflown by MABEL (e.g., ice, open water, 18 sea ice, or melt ponds) or to confirm the presence or absence of clouds. These images are 19 also available on the ICESat-2 website. The MABEL camera (Sony Nex7, with a 55 to 20 220 mm, f/4.5-6.6 telephoto lens) was mounted on the same optical bench as the MABEL 21 telescopes and shared the same portal in the aircraft. For the 2014 Alaska campaign, a 22 focal length of 210 mm was used for the duration of the campaign. The camera produced 23 6000 by 4000 pixel color images. At a nominal aircraft altitude of 20 km, each image 24 covers an approximately 2.25 by 1.5 km area, or approximately 3 m per pixel at sea level. 25 Jmages were taken every 3 seconds, which provided approximately 30% overlap between images. The images collected were not systematically georeferenced; however, they were 26 27 time-stamped based on MABEL instrument timing to provide a first-order assessment of 28 the surface that the lidar had surveyed. 29

30 2.3 Landsat 8 and WorldView-2 imagery

kelly brunt 6/14/2016 13:40 Deleted: E kelly brunt 6/14/2016 13:42 Deleted: . kelly brunt 5/18/2016 15:06

kelly brunt 6/15/2016 12:15 Deleted: measured

kelly brunt 5/16/2016 14:28

Data from the Landsat 8 Operational Land Imager (OLI) of the Bagley Icefield (Fig. 2b) 1 2 were used as an independent assessment of the depths of melt ponds surveyed by 3 MABEL. We applied spectrally based depthretrieval models to Landsat 8 imagery 4 (Moussavi et al., 2016; Moussavi, 2015; Pope et al., 2015), which were calibrated based 5 on data from supraglacial lakes in Greenland. We assessed the performance of OLI's 6 coastal blue, blue, green, red, and panchromatic channels in retrieving supraglacial lake 7 depths. Ultimately, the models establish a relationship between Landsat 8 top-ofatmosphere (TOA) comparing pre-drainage spectral reflectance values over the lakes 8 9 with a post-drainage digital elevation model (DEM), derived from WorldView-2 imagery 10 acquired from the Polar Geospatial Center at the University of Minnesota, using imageprocessing software (ERDAS). Our analysis indicated that for shallow lakes (depth

**28 2.4 Juneau Icefield GPS data**

29 Previous studies (Brunt et al., 2013; Brunt et al., 2014) have demonstrated that MABEL

30 precisely characterizes the ice-sheet surface when comparing MABEL-derived slope on

kelly brunt 6/15/2016 12:16 Deleted: on

kelly brunt 6/14/2016 13:46 Deleted: kelly brunt 6/16/2016 11:08 Deleted: ; Moussavi et al., 2014

kelly brunt 5/20/2016 11:09 Deleted: The models compare Landsat 8 kelly brunt 6/14/2016 13:50 Deleted: during pre-drainage

**kelly brunt 5/17/2016 17:38**

| Kelly brufit 5/20/2010 15.02                                   |    |
|----------------------------------------------------------------|----|
| Deleted: used to construct the Lower Tak
Glacier DEM | cu |
| kelly brunt 6/14/2016 13:51                                    |    |
| Deleted: ,                                                     |    |
| kelly brunt 6/14/2016 12:48                                    |    |
| Deleted: and                                                   |    |
|                                                                |    |

**kelly brunt 5/26/2016 15:52**

[revised manuscript text omitted]

**kelly brunt 5/18/2016 11:05**

kelly brunt 6/15/2016 12:27 **Deleted:** reflectivity,

kelly brunt 5/19/2016 16:01Deleted: ckelly brunt 5/19/2016 16:02Deleted: ckelly brunt 6/15/2016 12:28Deleted: , and for direct comparison with
ATLAS performance modelskelly brunt 6/15/2016 12:28Deleted: );kelly brunt 6/15/2016 12:29Deleted: theykelly brunt 5/19/2016 16:02Deleted: c

two campaigns, which include parameters such as the freshness of the most recent 1 2 snowfall, the dust content of the surface, the presence (or absence) of surface melt and 3 ponds, and the presence (or absence) of snow bridges that cover crevasses. Some 4 variation may also have been related to instrumentation issues, such as cleanliness of the 5 elements in the optics.

kelly brunt 6/15/2016 12:34 Deleted: sun angle,

| The MABEL signal-photon densities (Table 1) are generally Jower than those expected            |                           | kelly brunt 6/14/2016 13:53 |
|------------------------------------------------------------------------------------------------|---------------------------|-----------------------------|
| for ATLAS. Under similar conditions as the 2014 MABEL summer campaign and based                |                           | Deleted: less               |
| on performance models, we expect the strong beams of ATLAS to record 7.6 signal                |                           | kelly brunt 6/14/2016 13:53 |
| on performance models, we expect the strong beams of ATLAS to record $\frac{7.0}{7.0}$ signal  |                           | Deleted: that               |
| photons every shot (or $0.70$ m along track) over ice sheets and $0.5$ to $1.8$ signal photons | $\setminus$               | kelly brunt 6/14/2016 13:54 |
|                                                                                                |                           | Deleted: ,                  |
| every shot over the open ocean, dependent upon the state of the wind A. Martino, NASA          |                           | kelly brunt 6/1/2016 16:27  |
| GSFC, personal communication 2016). We note that for the Alaskan icefields, the                | $\langle \rangle \rangle$ | Deleted: 8.5                |
|                                                                                                |                           | kelly brunt 5/19/2016 16:02 |
| expected number of signal photons based on the performance model is probably too high,  | $\langle \rangle \rangle$ | Deleted: c                  |
| as the model uses an albedo of 0.9, which is more appropriate for ice with fresh snow or       |                           | kelly brunt 6/1/2016 14:38  |
|                                                                                                |                           | Deleted: 2.0                |
| the interior of Antarctica than for ice fields in Alaska in summer. Relative to the            |                           | kelly brunt 6/1/2016 16:10  |
| performance model at best ( i.e. using data from beam 50) the MABEL data used in this   |                           | Deleted:                    |
|                                                                                                |                           | kelly brunt 6/1/2016 14:38  |
| analysis suggest that the signal-photon densities were $\sim 72\%$ of the expected ATLAS       |                           | Deleted: 2014        |
| signal-photon densities over open ocean (with calm winds) and $\sim 40\%$ of the expected      |                           | kelly brunt 6/1/2016 16:17  |
| signal proton densities over open occan (with cann winds) and (7)/0 of the expected            |                           | Deleted: 65                 |
| ATLAS signal-photon densities over summer ice sheets                                           |                           | kelly brunt 6/1/2016 16:17  |
|                                                                                                |                           | Deleted: 14                 |

18 19

6

**20 3.2 Elevation bias and uncertainty**

21 We compared MABEL elevations to those based on the Juneau Icefield GPS array, 22 interpolated to the MABEL/GPS points of intersection (Fig. 3, blue solid points). The 23 mean offset, or bias, for the three points of intersection was  $3.2 \pm 0.1$  m. While this  $\ge 3$  m 24 instrument bias is larger than that of other airborne lidars, it is within the MABEL design 25 goals (algorithm development and error analysis), where instrument precision is more 26 critical to satellite algorithm development than absolute accuracy. Thus, while other 27 photon-counting systems are being used for change detection (e.g., Young et al., 2015), in its current configuration, MABEL is not suitable for time-series analysis of elevation 28 29 change, either independently or when integrated with other datasets.

| Deleted: less                                                                                                                                                              |
|----------------------------------------------------------------------------------------------------------------------------------------------------------------------------|
| kelly brunt 6/14/2016 13:53                                                                                                                                                |
| Deleted: that                                                                                                                                                              |
| kelly brunt 6/14/2016 13:54                                                                                                                                                |
| Deleted: ,                                                                                                                                                                 |
| kelly brunt 6/1/2016 16:27                                                                                                                                                 |
| Deleted: 8.5                                                                                                                                                               |
| kelly brunt 5/19/2016 16:02                                                                                                                                                |
| Deleted: c                                                                                                                                                                 |
| kelly brunt 6/1/2016 14:38                                                                                                                                                 |
| Deleted: 2.0                                                                                                                                                               |
| kelly brunt 6/1/2016 16:10                                                                                                                                                 |
| Deleted:                                                                                                                                                                   |
| kelly brunt 6/1/2016 14:38                                                                                                                                                 |
| Deleted: 2014                                                                                                                                                              |
| kelly brunt 6/1/2016 16:17                                                                                                                                                 |
| Deleted: 65                                                                                                                                                                |
| kelly brunt 6/1/2016 16:17                                                                                                                                                 |
| Deleted: 44                                                                                                                                                                |
| kelly brunt 6/15/2016 12:35                                                                                                                                                |
| Deleted: For ICESat-2 development
purposes, efforts are underway to merge data
from adjacent MABEL beams, which will
facilitate more direct MABEL to ATLAS |

kelly brunt 6/14/2016 13:55 Deleted: 08

comparisons.

kelly brunt 6/14/2016 13:56 Deleted: kelly brunt 6/14/2016 13:56 Deleted: , and ATLAS model validation

[revised manuscript text omitted]

kelly brunt 6/15/2016 12:36 Deleted: , kelly brunt 5/26/2016 15:30 Deleted: a kelly brunt 6/15/2016 12:36

kelly brunt 5/19/2016 16:02 Deleted: c kelly brunt 5/26/2016 15:32 Deleted: ( kelly brunt 5/26/2016 15:33 Deleted: – kelly brunt 5/26/2016 15:32 Deleted: ), kelly brunt 6/14/2016 13:59 Deleted: when

**kelly brunt 5/26/2016 15:33 Deleted: 3.3 - Local slope assessment for ice-sheet interiors kelly brunt 5/26/2016 15:33 Deleted: 4**

kelly brunt 6/14/2016 14:52 Deleted: indicated

kelly brunt 6/21/2016 12:42 Deleted: 500

kelly brunt 6/21/2016 12:41 Deleted: 56

Similarly, analysis of the individual beams in a different area of the Bagley Icefield 1 2 indicated that MABEL can determine the location of melt ponds. Fig. 6a shows stitched 3 MABEL images from crevasse and melt-pond fields on the Bagley Icefield; Fig. 6b 4 shows MABEL signal and background photons for a 1200 m range window that includes 5 the glacier surface; Fig. 6c shows both signal and background photon-count densities (per 125 shots, or ~2.5 m of along-track distance); and Fig. 6d shows MABEL signal photons, 6 7 indicating the location of two melt ponds, which are approximately 50 and 70 m in alongtrack length. The along-track slope of this crevasse and melt pond field, between 141.91° 8 9 and 141.93° W longitude in Fig. 6d, is 0.5°. A histogram of the signal photons associated 10 with the larger melt pond in Fig. 6d is provided in Fig. 7. This figure depicts how light at 532 and 1064 nm wavelengths interacts with the surface of the melt pond, and how the 11 melt pond affects the statistics of the 532 nm return signal. The FWHM for the 532 and 12 13 1064 nm return signal were 0.26 and 0.34 m, respectively. From Figs. 6 and 7 we observe 14 that while no distinct features corresponding to the bottoms of the melt ponds are visible, 15 an increased spread is apparent in the 532-nm histogram, likely associated with 16 volumetric scattering throughout the ponds. We applied spectrally based depth-retrieval 17 models to Landsat 8 imagery (Moussavi et al., 2016; Moussavi, 2015; Pope et al., 2015) 18 for an independent assessment of the depth of the melt-pond on the Bagley Icefield in 19 Fig. 6d. This analysis indicated that melt ponds in this region were approximately 2 m 20 deep. 21 Analysis of data from individual beams near the terminus of the Lower Taku Glacier

22 (Fig. 8) demonstrates MABEL performance in regions with steeper slopes. The slope in 23 this region is 4°, and is similar to slopes pear ice-sheet margins; this slope also 24 corresponds to the maximum slope angle used for ATLAS performance modeling over 25 ice-sheet margins (A. Martino, NASA GSFC, personal communication 2014). Fig. 8a 26 shows stitched MABEL camera images, which suggest a much rougher surface than that 27 of the low slope areas of interest on the Bagley Icefield examined in Fig. 6. Additionally, 28 the MABEL ice-surface signal near the terminus was slightly compromised due to 29 intermittent cloud cover, which attenuated the MABEL transmitted and/or received laser 30 pulses. Further, when cloud cover allows for only intermittent surface determination, the

kelly brunt 6/14/2016 14:01 Deleted: on kelly brunt 6/14/2016 14:02 Deleted: stretch

kelly brunt 6/21/2016 12:42 Deleted: 500

kelly brunt 6/21/2016 12:55 Deleted: a kelly brunt 6/21/2016 12:56 Deleted: is kelly brunt 6/6/2016 11:55 Deleted: 90 kelly brunt 6/6/2016 11:55 Deleted: 86 kelly brunt 6/21/2016 13:34 Deleted: 2 kelly brunt 6/21/2016 12:58 Deleted: location of the kelly brunt 6/21/2016 12:56 Deleted: the inset of kelly brunt 6/15/2016 12:38 Deleted: was generated to investigate kelly brunt 5/17/2016 17:30 **Deleted:** the penetration of kelly brunt 5/17/2016 17:30 Deleted: into the melt pond, kelly brunt 6/14/2016 14:03 Deleted: would kelly brunt 6/16/2016 11:10 Deleted: ; Moussavi et al., 2014 kelly brunt 6/15/2016 12:39 Deleted: provided kelly brunt 6/15/2016 12:39 Deleted: insight in to how kelly brunt 6/15/2016 12:39 Deleted: will operate kelly brunt 6/14/2016 14:04 Deleted: of kelly brunt 6/15/2016 12:40 Deleted: more consistent with kelly brunt 6/14/2016 14:04 Deleted: on kelly brunt 6/15/2016 12:40 Deleted: an kelly brunt 6/14/2016 14:04 Deleted: . A slope of 4° is also

[revised manuscript text omitted]

**kelly brunt 5/26/2016 15:33 Deleted: 5 Kelly brunt 5/26/2016 15:34 Deleted: for steeper glacial setting kelly brunt 5/26/2016 15:33 Deleted: s Kelly brunt 5/26/2016 16:26 Deleted: 5**

**kelly brunt 5/26/2016 16:25 Deleted: 2**

**kelly brunt 6/14/2016 14:13 Deleted: Similar kelly brunt 6/14/2016 14:13 Deleted: the methods of**

**kelly brunt 6/15/2016 12:54**

|   | Deleted: MABEL data from each beam were         |
|---|-------------------------------------------------|
|   | interpolated along track to a common time so    |
|   | that along-track elevations for each beam could |
|   | be used to calculate an across-track slope for  |
|   | each increment of along-track time              |
|   | kelly brunt 6/15/2016 12:54                     |
| J | Deleted: migrated                               |
|   | kelly brunt 6/15/2016 12:59                     |
| J | Deleted: Fig. 9a                                |
|   | kelly brunt 6/15/2016 12:59                     |
| J | Deleted: good agreement                         |
|   | kelly brunt 6/15/2016 12:59                     |
| V | Deleted: . Similarly,                           |
|   | kelly brunt 6/15/2016 12:59                     |
| V | Deleted: 9b                                     |
|   | kelly brunt 6/15/2016 12:59                     |

1 wavelengths). In order to relate the predicted performance of ATLAS with the measured 2 performance of MABEL, some common metric is necessary that accounts for as many of 3 the differences as is practicable. The signal-photon density is a metric to relate the 4 radiometery of the two instruments. Given that the signal-photon density is generally less 5 than that predicted for ATLAS, for a given background rate, the surface should be more 6 easily distinguished in ATLAS data. While in theory one could use the framework 7 developed for predicting ATLAS radiometric characteristics to make similar predictions 8 for MABEL and therefore use MABEL data to evaluate that framework, the efficiency or 9 radiometric throughput of MABEL has not been characterized well enough to do so. 10 Flight data (Brunt et al., 2014) show that for a given campaign, the measured signal-11 photon density of MABEL changes by tens of percent over relatively uniform ice sheet 12 interior. Similar changes are measured for the background rate, after consideration for 13 sun angle is taken into account. As such, the analysis presented here cannot be used to 14 guantitatively assess the likelihood that ATLAS will meet its measurement requirements 15 (or the mission science objectives). What we can say is that if the ATLAS signal-photon 16 density and signal-to-noise ratios are within 30% of its measurement requirements (and 17 thus mimics the MABEL performance documented in this study), ATLAS can be used to 18 measure surface slopes over both relatively flat ice-sheet interior conditions and steeper 19 glaciers such as the Lower Taku Glacier, and identify melt ponds. If ATLAS fully meets 20 its measurement requirements, we expect that the definition of small-scale surface 21 features such as crevasses and melt ponds will be correspondingly improved. 22 The result of this analysis indicates that the MABEL-derived local slope assessment, on a 23 relatively flat glacial surface and on a 90 m across-track length scale, is consistent with in

24 situ slope assessments made at spatial scales ranging from 50 to 150 m. For a planar 25 surface where slope is less than 1°, such as the interior of an ice sheet, we expect the local 26 slope measured by a GPS survey and MABEL to be similar over a wide range of spatial 27 scales. Any small differences observed between the two survey techniques would likely 28 reflect 1) the non-planarity of the surface and/or 2) the sensitivity of the results to small-29 scale slopes or roughness captured by one measurement technique and not the other. With 30 the good observed agreement between MABEL-derived and GPS-derived slope assessments over 50-150 m length scales (Fig. 9), we are confident that the ATLAS 90 m 31

 kelly brunt 6/14/2016 14:13

 kelly brunt 6/14/2016 14:14

 kelly brunt 6/14/2016 14:14

 kelly brunt 5/26/2016 16:27

 kelly brunt 6/15/2016 13:11

- 1 beam-spacing strategy will provide a non-aliased estimate of local slope for ice-sheet
- 2 interiors (<1°) over these spatial scales. This knowledge is necessary for accurate

3 assessments of ice-sheet surface-elevation change.

4 Based on our comparison with a WorldView-2-derived DEM of the Lower Taku Glacier, 5 MABEL can also provide valid estimates of across-track slope, even in steeper terrain. 6 Once migrated for GPS-derived ice-flow displacements, the southern part of the 7 MABEL-derived surface elevations are in good agreement with the DEM data, and the 8 slope comparison between MABEL-derived and DEM-derived across-track slopes had a 9 mean residual of 0.25°. This residual is larger than that reported over the Greenland Ice 10 Sheet (<0.05°) by Brunt et al. (2014), a difference that we attribute to errors associated 11 with the migration of the MABEL data (and the result of a flight line that was oblique to 12 the local direction of ice flow). Since the GPS array on the Lower Taku Glacier was not optimized to facilitate an across-track slope comparison similar to the comparison made 13 14 higher up on the Juneau Icefield (Figs. 3 and 9), we do not expect as close an agreement 15 between the two methods of estimating across-track slope. Figs. 5c and 6d suggest that the dense along-track sampling of MABEL is sufficient to 16 17 capture surface detail, including melt-pond information, from a single, static beam in 18 regions of low slope, consistent with that of an ice-sheet interior. Based on the continuous 19 nature of the surface return through the crevasse field, especially in the 1064 nm beam 20 (50) in Fig. 5c, we conclude that MABEL frequently retrieves a signal from the bottom of 21 crevasses. Further, Fig. 8b indicates that MABEL continues to provide surface detail in 22 regions of steeper slope, including the retrieval of the steep slopes of the crevasse walls 23 (*e.g.*, Figs 5c and 6d). 24 As previously noted, MABEL data used in this analysis had signal-photon densities that 25 are ~44% of the expected ATLAS signal-photon densities over summer ice sheets (A. 26 Martino, NASA GSFC, personal communication 2014). Therefore, we believe that the 27 level of detail that will be provided by ATLAS will be sufficient to determine local 28 surface characteristics, similar to those observed on the Lower Taku Glacier. Such

- 29 knowledge is critical to determining ice-sheet surface-elevation change, as features that
- 30 could compromise these calculations (such as deep crevasses), can move or advect with

**kelly brunt 5/26/2016 13:19 Deleted: robust kelly brunt 6/14/2016 14:14 Deleted: a kelly brunt 6/14/2016 14:14 Deleted: wide range (50 to 150 m) of**

| kelly brunt 6/21/2016 12:50      |
|----------------------------------|
| Deleted: 6c                      |
| kelly brunt 6/21/2016 12:50      |
| Deleted: 7d                      |
| kelly brunt 6/15/2016 13:12      |
| Deleted: provide                 |
| kelly brunt 6/15/2016 13:12      |
| Deleted: is generally retrieving |
| kelly brunt 6/15/2016 13:13      |
| Deleted: the                     |
| kelly brunt 6/21/2016 12:51      |
| Deleted: 6                       |
| kelly brunt 6/21/2016 12:51      |
| Deleted: 7d                      |
| kelly brunt 6/15/2016 13:13      |
| Deleted:                         |
| kelly brunt 6/14/2016 14:16      |
| Deleted: of                      |
| kelly brunt 6/14/2016 14:17      |
| Deleted: change                  |
| kelly brunt 6/14/2016 14:17      |
| Deleted: ,                       |
| kelly brunt 6/14/2016 14:17      |
| Deleted:                         |

1 ice-sheet flow or be bridged seasonally and must therefore be identifiable in the ATLAS

2 data.

The crevasse characterization we performed on the Bagley Icefield is qualitatively 3 confirmed using the camera imagery (Fig. 5a). However, it should be noted that we have 4 5 no means of quantitatively assessing the accuracy of MABEL-derived crevasse depths. 6 Crevasses on an ice-sheet surface have an influence on albedo (Pfeffer and Bretherton, 7 1987). This variation in reflectance is evident in Figs. 5b, 6b, and 6c, where MABEL 8 background photon counts, and the signal-to-noise ratios, change significantly. Changes 9 in MABEL background photon densities have also been used to detect leads in sea ice 10 (Kwok et al., 2014; Farrell et al., 2015). From Fig. 6c we note that the overall background 11 photon counts decrease significantly over the eastern region of this plot, which is 12 characterized by crevasses; however, this change is non-uniform. Background photon 13 counts drop steadily to nearly zero over the two melt ponds surveyed along this transect. 14 Penetration of 532 nm wavelength light into the surface, be it a melt pond or snow, is an 15 ongoing area of research for ICESat-2 algorithm development. MABEL geolocation 16 uncertainty, and the fact that the 1064 and 532 nm beams do not have coincident 17 footprints for more direct comparison (as the 1064 nm beams lead the 532 nm beams by 18 ~60 m), compromised our ability to further interrogate this topic with this dataset, as the 19 data could not be precisely co-registered spatially. Due to these limitations, a separate 20 campaign with a different photon-counting laser altimeter (with both a more accurate geopositioning system and coincident 1064 and 532 nm footprints) was deployed to 21 22 Thule, Greenland, in July and August 2015 (Brunt et al., 2015). Processing and analysis 23 of that dataset are still ongoing. 24 Analysis of MABEL data over small melt ponds on the Bagley Icefield in Alaska 25 provided a preliminary assessment of how green-wavelength photon-counting systems will interact with water on an ice surface. Based on the signal-photon elevations in Fig. 26 27 6d, and the histogram of the signal photons in Fig. 7, the total spread of the signal 28 photons, at a wavelength of 532 nm, is approximately 1.5 to 2 m. Further, analysis of

- 29 Landsat 8 and WorldView-2 imagery confirm that the melt ponds in this region are
- 30 | approximately 2 m deep. These results suggest that, while there isn't a distinct signal

**kelly brunt 6/14/2016 14:18 Deleted: associated with**

| 1                | kelly brunt 6/14/2016 14:18  |
|------------------|------------------------------|
|                  | Deleted: solar radiation and |
| 1                | kelly brunt 6/21/2016 12:52  |
|                  | Deleted: 6b                  |
|                  | kelly brunt 6/21/2016 12:52  |
| $\left  \right $ | Deleted: 7b                  |
| ì                | kelly brunt 6/21/2016 12:52  |
|                  | Deleted: 7c                  |
|                  |                              |

kelly brunt 6/14/2016 14:19 Deleted: . H

**kelly brunt 5/26/2016 16:33**

Deleted: The surface characterization of the Lower Taku Glacier is assessed using the camera imagery, WorldView-2 imagery, and a DEM derived from the WorldView-2 imagery (Fig. 8). Once the MABEL data have been migrated based on GPS ice-flow velocities, the southern part of the MABEL-derived surface elevations are in good agreement with the DEM data. However, the MABEL signal in this section is intermittent due to cloud cover. In the northern part of the MABEL data line, while the migration failed to improve surfaceelevation statistics, a generally continuous signal is detected, including melt ponds (Fig. 8b, inset). The slope comparison between MABEL-derived across-track slope and DEMderived across-track slope had a mean residual of 0.25°. This residual is larger than that reported over the Greenland Ice Sheet (<0.05°) by Brunt et al. (2014); we attribute this difference to errors associated with the migration of the MABEL data, given that the flight line was oblique to the local direction of ice flow. Since the GPS array on the Lower Taku Glacier was not optimized to facilitate an across-track slope comparison similar to the comparison made higher up on the Juneau Icefield (Figs. 3 and 5), we do not expect good agreement between the two methods of estimating across-track slope.

kelly brunt 6/21/2016 13:02 Deleted: in the inset

return from a melt-pond bottom, the 532 nm MABEL beam may be sampling the entire
 melt-pond water column. The 1064 nm MABEL beam shows evidence of a secondary
 return 1.5 m below the main signal return, due to unintended secondary pulses from the
 MABEL laser that occur under some operational conditions, and is likely not due to melt pond bottom returns.

6 Based on the surface characterization results of MABEL data from the Juneau and 7 Bagley icefields, and the dense, six-beam sampling strategy of ATLAS, we are confident 8 that ICESat-2 will contribute significantly to glacier studies at local and regional scales 9 and in polar and mid-latitudes. While previous studies using satellite laser altimetry have 10 investigated the vertical dimension of rifts in the ice sheet (e.g., Fricker et al., 2005), 11 those studies have been limited to major ice-shelf rift systems, as opposed to smaller-12 scale crevasses. The 0.70 m along-track sampling density of each individual ATLAS 13 beam is well suited for similar vertical dimension studies, but at finer length-scales, such 14 as those associated with alpine glacier crevasse fields.

kelly brunt 6/15/2016 13:15 Deleted: feel

kelly brunt 5/19/2016 16:03

15

**16 5 Conclusions**

17 Knowledge of local slope and local surface character are required to accurately determine 18 ice-sheet surface-elevation change. The ATLAS beam geometry includes pairs of beams 19 separated at 90 m across track to enable the determination of local slope in one pass, and therefore to enable the determination of ice-sheet surface-elevation change in just two 20 21 passes. Based on the analysis of MABEL, ground-based GPS data, and the resultant 22 surface gradient comparison (SGC), we conclude that the ATLAS 90 m beam-spacing 23 strategy will provide a valid assessment of local slope that is consistent with the slope of 24 an ice-sheet interior (<1°) on 50 to 150 m length scales. The density of along-track 25 photon-counting lidar data is sufficient to characterize the ice-sheet surface in detail, 26 including small-scale features such as crevasses and melt ponds. This information is 27 required for accurate determination of ice-sheet surface-elevation change. The dense 28 along-track sampling interval and narrow across-track beam spacing of ATLAS will 29 provide a level of detail of mountain glaciers that has previously not been achieved from 30 satellite laser altimetry. While studies of mountain glaciers stand to benefit greatly from

| kelly brunt 6/14/2016 14:20 |
|-----------------------------|
| Deleted: ,                  |
| kelly brunt 6/14/2016 14:20 |
| Deleted: ,                  |
| kelly brunt 6/14/2016 14:21 |
| Deleted: and                |
| kelly brunt 5/26/2016 13:19 |
| Deleted: robust             |
|                             |

kelly brunt 6/17/2016 15:52 Deleted: also

[revised manuscript text omitted]